# Unit Ball Model for Embedding Hierarchical Structures in the Complex Hyperbolic Space

## Abstract

Learning the representation of data with hierarchical structures in the hyperbolic space attracts increasing attention in recent years. Due to the constant negative curvature, the hyperbolic space resembles tree metrics and captures the tree-like properties naturally, which enables the hyperbolic embeddings to improve over traditional Euclidean models. However, many real-world hierarchically structured data such as taxonomies and multitree networks have varying local structures and they are not trees, thus they do not ubiquitously match the constant curvature property of the hyperbolic space. To address this limitation of hyperbolic embeddings, we explore the complex hyperbolic space, which has the variable negative curvature, for representation learning. Specifically, we propose to learn the embeddings of hierarchically structured data in the unit ball model of the complex hyperbolic space. The unit ball model based embeddings have a more powerful representation capacity to capture a variety of hierarchical structures. Through experiments on synthetic and real-world data, we show that our approach improves over the hyperbolic embedding models significantly.

## 1 Introduction

Representation learning of data with hierarchical structures is an important machine learning task with many applications, such as taxonomy induction (Fu et al., 2014) and hypernymy detection (Shwartz et al., 2016). In recent years, the hyperbolic embeddings (Nickel and Kiela, 2017, 2018) have been proposed to improve the traditional Euclidean embedding models (Nickel et al., 2011; Bordes et al., 2013). The constant negative curvature of the hyperbolic space produces several manifestations, where the most desirable property for representation learning is that the hyperbolic space can be regarded as a continuous approximation to trees (Krioukov et al., 2010). The hyperbolic space is capable of embedding any finite tree while preserving the distances approximately (Gromov, 1987). As a result of the tree-like properties, the hyperbolic space is more suitable to embed hierarchically structured data than Euclidean space.

However, the real-world hierarchically structured data are usually not trees since they can have varying local structures while being tree-like globally. For example, although the taxonomies such as WordNet (Miller, 1995) and YAGO (Suchanek et al., 2007) have underlying hierarchical structures, they contain many $1$-$n$ (1 child links to multiple parents) cases and multitree structures (Griggs et al., 2012), which are much more complicated than trees. Thus, the general hierarchically structured data cannot ubiquitously match the constant negative curvature property of the hyperbolic space.

To address the challenge, in this paper, we present a new approach to learning the embeddings of hierarchically structured data. Specifically, we embed the data with hierarchical structures into the unit ball model of the complex hyperbolic space. The unit ball model is a projective geometry based model to identify the complex hyperbolic space. One of the main differences between the complex

and the real hyperbolic space is that the curvature is no longer constant in the complex hyperbolic space. Instead, it has the variable negative curvature. In practice, the variable negative curvature makes the unit ball model based embeddings more flexible in handling varying structures while the tree-like properties retain the superiority in hierarchies.

For empirical evaluation, we first compare our approach with the hyperbolic embedding methods on tree structures to show that the complex hyperbolic space maintains the tree-like properties. Then we evaluate our approach and the baselines on various hierarchically structured data, including synthetic graphs and real-world taxonomies. The experimental results demonstrate the advantages of our approach. To summarize, our work has the following main contributions:

1. We present a novel embedding approach, which takes advantage of the variable negative curvature of the complex hyperbolic space, to handle data with complicated and various hierarchical structures. To the best of our knowledge, our work is the first to propose complex hyperbolic embeddings.

2. We introduce the embedding algorithm in the unit ball model of the complex hyperbolic space. We formulate the learning and Riemannian optimization in the unit ball model.

3. We evaluate our approach with experiments on an extensive range of synthetic and real-world data and show the remarkable improvements of our approach.

## 2 Related work

**Hyperbolic embeddings.** Hyperbolic embedding methods have become the leading approach for representation learning of hierarchical structures. (Nickel and Kiela, 2017) learned the representations of hierarchical graphs in the Poncaré ball model of the hyperbolic space and obtained high-quality embeddings for taxonomies. (Ganea et al., 2018a) introduced the hyperbolic entailment cones to formally define the partial ordering relation. (Nickel and Kiela, 2018) proposed to learn the embeddings in the hyperboloid model (also known as the Lorentz model) of the hyperbolic space to avoid the numerical instabilities of the Poncaré ball model. These methods learned the hyperbolic embeddings by Riemannian optimization (Bonnabel, 2013), which was further improved by the Riemannian adaptive optimization (Bécigneul and Ganea, 2019). Additionally, (Yu and Sa, 2019) used an integer-based tiling to solve the numerical instabilities in the hyperbolic embeddings.

Another branch of study (Sala et al., 2018; Sonthalia and Gilbert, 2020) learned the hyperbolic embeddings through combinatorial construction. Instead of optimizing the soft-ranking loss by Riemannian SGD to preserve the hierarchical relationships as in (Nickel and Kiela, 2017, 2018), the construction-based methods minimize the reconstruction distortion and focus on the graph reconstruction task. Remarkably, TreeRep (Sonthalia and Gilbert, 2020) can exactly recover the original tree structure when the given graph is a tree. However, both the optimization-based and construction-based hyperbolic embeddings suffer from the limitation in hierarchical graphs with varying local structures. To tackle the challenge, (Gu et al., 2019) extended the construction-based method by jointly learning the curvature and the embeddings of data in a product manifold. Although it can provide a better representation than a single space with constant curvature, it is impractical to search for the best manifold combination among enormous combinations for each new structure.

Note that our complex hyperbolic embedding model is different from the hyperbolic embedding methods (Nickel and Kiela, 2017, 2018) or the product manifold embeddings (Gu et al., 2019) since the geometrical spaces are typically of different characteristics. The $n$-dimensional ($n$-d) complex hyperbolic space is not simply the $2n$-d hyperbolic space or the product of two $n$-d hyperbolic spaces. Section 3 will show that their geometries differ markedly.

Motivated by the promising results of previous works, extensions to the multi-relational graph hyperbolic embeddings (Balazevic et al., 2019; Chami et al., 2020; Sun et al., 2020) and hyperbolic neural networks (Ganea et al., 2018b; Gülçehre et al., 2019; Liu et al., 2019; Chami et al., 2019; Dai et al., 2021; Shimizu et al., 2021) were explored. Notably, (Chami et al., 2019, 2020) leverages the trainable curvature to compensate for the disparity between the actual data structures and the constant-curvature hyperbolic space, where each layer in the graph neural network or each relation in the multi-relational graph has its own curvature parameterization. Since we only focus on the single-relation graph embeddings and taxonomy embeddings in this work, we do not evaluate the multi-relational knowledge graph embedding models or the neural networks in our tasks.

**Complex embeddings.** The traditional knowledge graph embeddings were learned in the real Euclidean space (Nickel et al., 2011; Bordes et al., 2013; Yang et al., 2015) and were used for knowledge graph inference and reasoning. In recent years, several works suggested utilizing the complex Euclidean space for inferring more relation patterns, such as ComplEx (Trouillon et al., 2016) and RotatE (Sun et al., 2019). The computation operations and transformations in the complex space have been demonstrated to be effective in the knowledge graph embeddings. The success of the complex embeddings reveals the potential of the complex space and inspires us to explore the complex hyperbolic space.

## 3 Preliminaries

### 3.1 Curvature

Before introducing the hyperbolic geometry and the complex hyperbolic geometry, we need to give the definition of *curvature*, which describes the curve of Riemannian manifolds and controls the rate of geodesic deviation. In this paper, *curvature* refers to the *sectional curvature*.

**Definition 1** (Curvature). *Given a Riemannian manifold and two linearly independent tangent vectors at the same point,* $\mathbf{u}$ *and* $\mathbf{v}$*, the **(sectional) curvature** is defined as*

$$K(\mathbf{u}, \mathbf{v}) = \frac{\langle R(\mathbf{u}, \mathbf{v})\mathbf{v}, \mathbf{u} \rangle}{\langle \mathbf{u}, \mathbf{u} \rangle \langle \mathbf{v}, \mathbf{v} \rangle - \langle \mathbf{u}, \mathbf{v} \rangle^2},$$

*where* $R$ *is the Riemann curvature tensor, defined by the convention* $R(\mathbf{u}, \mathbf{v})\mathbf{w} = \nabla_{\mathbf{u}}\nabla_{\mathbf{v}}\mathbf{w} - \nabla_{\mathbf{v}}\nabla_{\mathbf{u}}\mathbf{w} - \nabla_{[\mathbf{u}, \mathbf{v}]}\mathbf{w}$.

### 3.2 Hyperbolic geometry

Hyperbolic space[1] is a homogeneous space with constant negative curvature. Here *constant* means constant both at all points and in all pairs of directions. In the hyperbolic space $\mathbb{H}_{\mathbb{R}}^n(K)$ of dimension $n$ and curvature $K < 0$, the volume of a ball grows exponentially with its radius $\rho$:

$$vol(B_{\mathbb{H}_{\mathbb{R}}^n(K)}(\rho)) \sim e^{\sqrt{-K}(n-1)\rho}. \tag{1}$$

Contrastively, in the Euclidean space $\mathbb{E}^n$, the curvature is $0$ and the volume of a ball grows polynomially with its radius:

$$vol(B_{\mathbb{E}^n}(\rho)) = \frac{\pi^{\frac{n}{2}}}{\Gamma(\frac{n}{2})}\rho^n \sim \rho^n. \tag{2}$$

The exponential volume growth rate enables the hyperbolic space to have powerful representation capability for tree structures since the number of nodes grows exponentially with the depth in a tree, while the Euclidean space is too flat and narrow to embed trees.

### 3.3 Complex hyperbolic geometry

Complex hyperbolic space is a homogeneous geometry of variable negative curvature. Its ambient Hermitian vector space $\mathbb{C}^{n,1}$ is the complex Euclidean space $\mathbb{C}^{n+1}$ endowed with a Hermitian form $\langle\!\langle \mathbf{z}, \mathbf{w} \rangle\!\rangle$, where $\mathbf{z}, \mathbf{w} \in \mathbb{C}^{n+1}$. Then the Hermitian space $\mathbb{C}^{n,1}$ can be divided into three subsets: $V_- = \{\mathbf{z} \in \mathbb{C}^{n,1} | \langle\!\langle \mathbf{z}, \mathbf{z} \rangle\!\rangle < 0\}$, $V_0 = \{\mathbf{z} \in \mathbb{C}^{n,1} - \{\mathbf{0}\} | \langle\!\langle \mathbf{z}, \mathbf{z} \rangle\!\rangle = 0\}$, and $V_+ = \{\mathbf{z} \in \mathbb{C}^{n,1} | \langle\!\langle \mathbf{z}, \mathbf{z} \rangle\!\rangle > 0\}$. Let $\mathbb{P}$ be a projection map $\mathbb{P} : \mathbb{C}^{n,1} - \{z_{n+1} = 0\} \to \mathbb{C}^n$, i.e.,

$$\mathbb{P} : \begin{bmatrix} z_1 \\ \dots \\ z_{n+1} \end{bmatrix} \mapsto \begin{bmatrix} z_1/z_{n+1} \\ \dots \\ z_n/z_{n+1} \end{bmatrix}, \text{where } z_{n+1} \neq 0. \tag{3}$$

Then the complex hyperbolic space $\mathbb{H}_{\mathbb{C}}^n$ and its boundary $\partial\mathbb{H}_{\mathbb{C}}^n$ are defined using the projectivization:

$$\mathbb{H}_{\mathbb{C}}^n = \mathbb{P}V_-, \qquad \partial\mathbb{H}_{\mathbb{C}}^n = \mathbb{P}V_0. \tag{4}$$

The curvature of the complex hyperbolic space is summarized by (Goldman, 1999) as follows:

---

[1] In this paper, we use *hyperbolic space* to refer to real hyperbolic space and *hyperbolic embeddings* to refer to real hyperbolic embeddings for avoiding wordiness.

**Theorem 1.** *The curvature is not constant in $\mathbb{H}_{\mathbb{C}}^n$. It is pinched between $-1$ (in the directions of complex projective lines) and $-1/4$ (in the directions of totally real planes).*

We leave the full proof in Appendix A. The non-constant curvature, which we expect to be favorable for embedding various hierarchical structures, is one of the main differences between $\mathbb{H}_{\mathbb{C}}^n$ and the real hyperbolic space $\mathbb{H}_{\mathbb{R}}^n$.

The complex hyperbolic space also has the tree-like exponential volume growth property. The volume of a ball with radius $\rho$ in $\mathbb{H}_{\mathbb{C}}^n$ is given by

$$vol(B_{\mathbb{H}_{\mathbb{C}}^n}(\rho)) = \frac{8^n \sigma_{2n-1}}{2n} \sinh^{2n}(\rho/2) \sim \frac{8^n \sigma_{2n-1}}{2n} e^{n\rho}, \tag{5}$$

where $\sigma_{2n-1} = 2\pi^n/n!$ is the Euclidean volume of the unit sphere $S^{2n-1} \in \mathbb{C}^n$.

From the properties of the complex hyperbolic geometry, we expect that the complex hyperbolic space can naturally handle data with diverse local structures in virtue of the variable curvature as presented in Theorem 1 while preserving the tree-like properties as shown in Eq. (5).

# 4 Unit ball embeddings

We propose to embed the hierarchically structured data into the unit ball model of the complex hyperbolic space. In this section, We introduce our approach in detail.

## 4.1 The unit ball model

The unit ball model is one model used to identify the complex hyperbolic space, which can be derived via the projective geometry (Goldman, 1999). We now provide the derivation sketch.

Take the Hermitian form of $\mathbb{C}^{n,1}$ in Section 3.3 to be a standard Hermitian form:

$$\langle\!\langle \mathbf{z}, \mathbf{w} \rangle\!\rangle = z_1 \overline{w_1} + \cdots + z_n \overline{w_n} - z_{n+1} \overline{w_{n+1}}, \tag{6}$$

where $\overline{w}$ is the conjugate of $w$. Take $z_{n+1} = 1$ in the projection map $\mathbb{P}$ in Eq. (3), then from Eq. (4), we can derive the formula of the unit ball model:

$$\mathcal{B}_{\mathbb{C}}^n = \{(z_1, \cdots, z_n, 1) | |z_1|^2 + \cdots + |z_n|^2 < 1\}, \tag{7}$$

where $|\cdot|$ is the Euclidean norm.

The metric on $\mathcal{B}_{\mathbb{C}}^n$ is Bergman metric, which takes the formula below in 2-d case:

$$ds^2 = \frac{-4}{\langle\!\langle \mathbf{z}, \mathbf{z} \rangle\!\rangle^2} \det \begin{bmatrix} \langle\!\langle \mathbf{z}, \mathbf{z} \rangle\!\rangle & \langle\!\langle d\mathbf{z}, \mathbf{z} \rangle\!\rangle \\ \langle\!\langle \mathbf{z}, d\mathbf{z} \rangle\!\rangle & \langle\!\langle d\mathbf{z}, d\mathbf{z} \rangle\!\rangle \end{bmatrix}. \tag{8}$$

The distance function on $\mathcal{B}_{\mathbb{C}}^n$ is given by

$$d_{\mathcal{B}_{\mathbb{C}}^n}(\mathbf{z}, \mathbf{w}) = arcosh(2 \frac{\langle\!\langle \mathbf{z}, \mathbf{w} \rangle\!\rangle \langle\!\langle \mathbf{w}, \mathbf{z} \rangle\!\rangle}{\langle\!\langle \mathbf{z}, \mathbf{z} \rangle\!\rangle \langle\!\langle \mathbf{w}, \mathbf{w} \rangle\!\rangle} - 1), \tag{9}$$

where the Hermitian form $\langle\!\langle \mathbf{z}, \mathbf{w} \rangle\!\rangle$ is defined in Eq. (6).

## 4.2 Embeddings in the unit ball model

Given the hierarchical data containing a set of nodes $X = \{x_p\}_{p=1}^m$ and a set of edges $E = \{(x_p, x_q) | x_p, x_q \in X\}$, we aim to learn the embeddings of the nodes $\mathbf{Z} = \{\mathbf{z}_p\}_{p=1}^m$, where $\mathbf{z}_p \in \mathcal{B}_{\mathbb{C}}^n$.

The objective of the embeddings is to recover the structures of input data, including the distances between the nodes as well as the partial order in the hierarchies. Here we adopt the soft ranking loss used in the Poincaré ball embeddings (Nickel and Kiela, 2017) and the hyperboloid embeddings (Nickel and Kiela, 2018), which aims at preserving the hierarchical relationships among nodes:

$$L = \sum_{(x_p, x_q) \in E} \log \frac{e^{-d_{\mathcal{B}_{\mathbb{C}}^n}(\mathbf{z}_p, \mathbf{z}_q)}}{\sum_{x_k \in \mathcal{N}(x_p)} e^{-d_{\mathcal{B}_{\mathbb{C}}^n}(\mathbf{z}_p, \mathbf{z}_k)}}, \tag{10}$$

---

**Algorithm 1** RSGD of the unit ball embeddings.

---

**Input:** initialization $\mathbf{z}^{(0)}$, number of iterations $T$, learning rates $\{\eta^{(t)}\}_{t=1}^{T}$.
**for** $t = 1$ **to** $T$ **do**
    Compute $\frac{\partial d_{\mathcal{B}_{\mathbb{C}}^n}}{\partial \mathbf{x}}$ and $\frac{\partial d_{\mathcal{B}_{\mathbb{C}}^n}}{\partial \mathbf{y}}$ by Eqs. (14) and (15).
    Compute $\nabla_E L(\mathbf{z})$ and $\nabla_R L(\mathbf{z})$ by Eq. (13).
    Update $\mathbf{z}^{(t)}$ by Eq. (17).
**end for**

---

where $\mathcal{N}(x_p) = \{x_k : (x_p, x_k) \notin E_\mathcal{T}\} \cup \{x_p\}$ is the set of negative examples for $x_p$ together with $x_p$. $d_{\mathcal{B}_{\mathbb{C}}^n}$ is the distance function in the unit ball model given in Eq. (9). The minimization of $L$ makes the connected nodes closer in the embedding space than those with no observed edges.

Note that instead of manually setting the curvature of the learning space or training the curvature as extra parameters, we learn the embeddings directly in the complex hyperbolic space, where the curvature is variable. The learned embeddings are located in different submanifolds of the unit ball model, whose curvatures are different.

### 4.3 Riemannian optimization in the unit ball model

We learn the embeddings $\mathbf{Z} = \{\mathbf{z}_p\}_{p=1}^{m}$ through solving the optimization problem with constraint:

$$\mathbf{Z} \leftarrow \arg\min_{\mathbf{Z}} L \qquad s.t. \forall \mathbf{z}_p \in \mathbf{Z}, \mathbf{z}_p \in \mathcal{B}_{\mathbb{C}}^n. \tag{11}$$

For the optimization problems in Riemannian manifolds, (Bonnabel, 2013) presented the Riemannian stochastic gradient descent (RSGD) algorithm, which we employ to optimize Eq. (11). To update an embedding $\mathbf{z} \in \mathcal{B}_{\mathbb{C}}^n$,[2] we need to obtain its Riemannian gradient $\nabla_R$. Specifically, denote $\mathcal{T}_{\mathbf{z}}\mathcal{B}_{\mathbb{C}}^n$ as the tangent space of $\mathbf{z}$, then the embedding is updated at the $t$-th iteration by

$$\mathbf{z}^{(t)} \leftarrow \mathbf{z}^{(t-1)} - \eta^{(t)} \nabla_R L(\mathbf{z}), \tag{12}$$

where $\eta^{(t)}$ is the learning rate at the $t$-th iteration and $\nabla_R L(\mathbf{z}) \in \mathcal{T}_{\mathbf{z}}\mathcal{B}_{\mathbb{C}}^n$ is the Riemannian gradient of $L(\mathbf{z})$. Then the Riemannian gradient $\nabla_R$ can be derived from rescaling the Euclidean gradient $\nabla_E$ with the inverse of the metric tensor $ds^2$ and applying the chain rule of differential functions:

$$\nabla_R L(\mathbf{z}) = \frac{1}{ds^2} \nabla_E L(\mathbf{z}) = \frac{1}{ds^2} \frac{\partial L(\mathbf{z})}{\partial d_{\mathcal{B}_{\mathbb{C}}^n}(\mathbf{z}, \mathbf{w})} \nabla_E d_{\mathcal{B}_{\mathbb{C}}^n}(\mathbf{z}, \mathbf{w}), \tag{13}$$

where $ds^2$ is in Eq. (8) and $\frac{\partial L(\mathbf{z})}{\partial d_{\mathcal{B}_{\mathbb{C}}^n}(\mathbf{z}, \mathbf{w})}$ is trivial to compute from Eq. (10).

In practical training, we implement and compute the complex hyperbolic embedding as its real part and imaginary part, i.e., $\mathbf{z} = \mathbf{x} + i\mathbf{y}$, where $i$ represents the *imaginary unit*, i.e., $i^2 = -1$. In order to get the gradient of the distance function $\nabla_E d_{\mathcal{B}_{\mathbb{C}}^n}(\mathbf{z}, \mathbf{w})$ in Eq. (13), we get the partial derivative with regard to the real part and the imaginary part, i.e., $\nabla_E d_{\mathcal{B}_{\mathbb{C}}^n}(\mathbf{z}, \mathbf{w}) = \frac{\partial d_{\mathcal{B}_{\mathbb{C}}^n}(\mathbf{z}, \mathbf{w})}{\partial \mathbf{x}} + i \frac{\partial d_{\mathcal{B}_{\mathbb{C}}^n}(\mathbf{z}, \mathbf{w})}{\partial \mathbf{y}}$.

The partial derivatives of the unit ball model distance take the following formulas:

$$\frac{\partial d_{\mathcal{B}_{\mathbb{C}}^n}}{\partial \mathbf{x}} = \frac{4}{\sqrt{p^2 - 1}} \left( \frac{Re(\langle\!\langle \mathbf{z}, \mathbf{w} \rangle\!\rangle \mathbf{w})}{\langle\!\langle \mathbf{z}, \mathbf{z} \rangle\!\rangle \langle\!\langle \mathbf{w}, \mathbf{w} \rangle\!\rangle} - \frac{\langle\!\langle \mathbf{z}, \mathbf{w} \rangle\!\rangle \langle\!\langle \mathbf{w}, \mathbf{z} \rangle\!\rangle \mathbf{x}}{\langle\!\langle \mathbf{z}, \mathbf{z} \rangle\!\rangle^2 \langle\!\langle \mathbf{w}, \mathbf{w} \rangle\!\rangle} \right), \tag{14}$$

$$\frac{\partial d_{\mathcal{B}_{\mathbb{C}}^n}}{\partial \mathbf{y}} = \frac{4}{\sqrt{p^2 - 1}} \left( \frac{Im(\langle\!\langle \mathbf{z}, \mathbf{w} \rangle\!\rangle \mathbf{w})}{\langle\!\langle \mathbf{z}, \mathbf{z} \rangle\!\rangle \langle\!\langle \mathbf{w}, \mathbf{w} \rangle\!\rangle} - \frac{\langle\!\langle \mathbf{z}, \mathbf{w} \rangle\!\rangle \langle\!\langle \mathbf{w}, \mathbf{z} \rangle\!\rangle \mathbf{y}}{\langle\!\langle \mathbf{z}, \mathbf{z} \rangle\!\rangle^2 \langle\!\langle \mathbf{w}, \mathbf{w} \rangle\!\rangle} \right), \tag{15}$$

where $p = \cosh(d_{\mathcal{B}_{\mathbb{C}}^n}(\mathbf{z}, \mathbf{w}))$, $Re(\cdot)$ and $Im(\cdot)$ denote the real and the imaginary part respectively. The full derivation of Eqs. (14) and (15) is given in Appendix B.

Since the embedding $\mathbf{z}$ should be constrained within the unit ball model, we apply the same projection strategy as (Nickel and Kiela, 2017) via a small constant $\varepsilon$:

$$proj(\mathbf{z}) = \begin{cases} \mathbf{z}/(|\mathbf{z}| - \varepsilon) & \text{if } |\mathbf{z}| \geq 1, \\ \mathbf{z} & \text{otherwise.} \end{cases} \tag{16}$$

---

[2]Here we omit the subscript of $\mathbf{z}_p$ for concision.

Table 1: The real-world datasets statistics.

|  | ICD10 | YAGO3-wikiObjects | WordNet-noun |
|---|---|---|---|
| Nodes | 19,155 | 17,375 | 82,115 |
| Edges | 78,357 | 153,643 | 743,086 |
| Depth | 6 | 16 | 20 |
| Training edges | 70,521 | 138,277 | 668,776 |
| Valid/Test edges | 3,918 | 7,683 | 37,155 |
| $\delta$-hyperbolicity | 0.0 | 1.0 | 0.5 |

To sum up, the update of $\mathbf{z}$ at the $t$-th iteration is

$$\mathbf{z}^{(t)} \leftarrow proj\big(\mathbf{z}^{(t-1)} - \eta^{(t)}\nabla_R L(\mathbf{z})\big) = proj\big(\mathbf{z}^{(t-1)} - \eta^{(t)}\frac{1}{ds^2}\nabla_E L(\mathbf{z})\big). \qquad (17)$$

The RSGD steps of the unit ball embeddings are presented in Algorithm 1.

## 5 Experiments

In this section, we evaluate the performances of our approach on tree structures and various hierarchical structures, including synthetic graphs and real-world taxonomies. We focus on the graph reconstruction and link prediction tasks. For more experiments, please refer to Appendix D.

### 5.1 Experimental settings

#### 5.1.1 Data

We use synthetic and real-world data that exhibit underlying hierarchical structures to evaluate our approach. The details are as follows.

**Synthetic.** We generate various balanced trees and compressed graphs using NetworkX package (Hagberg et al., 2008).[3] For **balanced trees**, we generate the balanced tree with degree $r$ and depth $h$. For **compressed graphs**, we generate $k$ random trees on $m$ nodes and then aggregate their edges to form a graph. Some examples of the synthetic data are given in Appendix D.1.

**ICD10.** The 10-th revision of International Statistical Classification of Diseases and Related Health Problems (ICD10)[4] (Brämer, 1988) is a medical classification list provided by the World Health Organization. The classification list forms a tree structure. We construct its full transitive closure as the ICD10 dataset.

**YAGO3-wikiObjects.** YAGO3[5] (Mahdisoltani et al., 2015) is a huge semantic knowledge base. It provides a taxonomy derived from Wikipedia and WordNet. We extract the Wikipedia concepts and entities that are descendants of $\langle wikicat\_Objects \rangle$ as well as the hypernymy edges among them. We compute the transitive closure of the sampled taxonomy to construct the YAGO3-wikiObjects dataset.

**WordNet-noun.** WordNet[6] (Miller, 1995) is a large lexical database. The hypernymy relation among all nouns forms a noun hierarchy. We use its full transitive closure as the WordNet-noun dataset.

For each real-world dataset, we randomly split the edges into train-validation-test sets with the ratio 90%:5%:5%. We make sure that any node in the validation and test sets must occur in the training set since otherwise, it cannot be predicted. But the edges in the validation and test sets do not occur in the training set since they are disjoint. We provide the statistics of the real-world datasets in Table 1. The Gromov's $\delta$-hyperbolicity (Gromov, 1987) measures the tree-likeness of graphs (refer to Appendix C for definition). The lower $\delta$ corresponds to the more tree-like graph and trees have 0 $\delta$-hyperbolicity.

#### 5.1.2 Tasks

We evaluate the following two tasks:

---

[3]https://networkx.org/documentation/stable/reference/generators.html
[4]https://www.who.int/standards/classifications/classification-of-diseases
[5]https://yago-knowledge.org/
[6]https://wordnet.princeton.edu/

**Graph reconstruction.** We train the embeddings of the full data and then reconstruct it from the embeddings. The task evaluates representation capacity.

**Link prediction.** We train the embeddings on the training set and predict the edges in the test set. The task evaluates generalization performance.

### 5.1.3 Baselines

We compare our approach **UnitBall** to the following methods: the sate-of-the-art combinatorial construction-based hyperbolic embedding method **TreeRep** (Sonthalia and Gilbert, 2020), the optimization-based hyperbolic embeddings in the **Poincaré** ball model (Nickel and Kiela, 2017) and the **Hyperboloid** model (Nickel and Kiela, 2018), the simple **Euclidean** embedding model using the same loss function with (Nickel and Kiela, 2017, 2018). Recall that we use the same loss function with Poincaré and Hyperboloid but learn in the unit ball model. Therefore, the comparisons among UnitBall, Poincaré, Hyperboloid, and Euclidean reveal the representation capacities of different geometrical models in different spaces.

For the baselines, we use their public codes to train the embeddings. For all methods, the hyperparameters are tuned on each validation set for link prediction task and on balanced tree-(15,3) for graph reconstruction task. The hardware information is given in Appendix D.2 and the hyperparameters are listed in Appendix D.3. In all experiments, we report the mean results over $5$ running executions. The code of our approach will be publicly available after the publishing of the paper.

### 5.1.4 Evaluation

We use the mean average precision (**MAP**), mean reciprocal rank (**MRR**), and **Hits@N** as our evaluation metrics, which are widely used for evaluating ranking and link prediction. The details of prediction steps and the evaluation metrics are given in Appendix D.4.

The $n$-d complex hyperbolic embeddings have around double parameters of the $n$-d real embeddings since the $n$-d complex hyperbolic vectors have $n$-d real part and $n$-d imaginary part. For a fair comparison, in each experimental setting, we compare our $n$-d complex hyperbolic embeddings of UnitBall against the $2n$-d embeddings of the baselines. The results will also demonstrate that the $n$-d complex hyperbolic space is not simply the $2n$-d hyperbolic space, they have different capacities.

## 5.2 Graph reconstruction

### 5.2.1 Results on balanced trees

To compare the representation capacities of UnitBall and the hyperbolic embedding models for the tree structures, we first evaluate the graph reconstruction task on the synthetic balanced trees. A balanced tree-$(r, h)$ has degree $r$ and depth $h$, so it has $r^0 + \cdots + r^d$ nodes and $r^0 + \cdots + r^d - 1$ edges. The $\delta$-hyperbolicity of any balanced tree is $0$. We embed the balanced trees into 20-d hyperbolic space for the baselines and 10-d complex hyperbolic space for UnitBall.

Figure 1 presents the MAP and Hits@3 scores with varying $r$ and $h$. We see that when the tree is in small scale, e.g., $(r, h) = (15, 3), (10, 2), (10, 3)$, all methods have very good performances, demonstrating the expected powerful capacities of hyperbolic geometry and complex hyperbolic geometry on tree structures. However, when the breadth or the depth increases, the performances of Poincaré and Hyperboloid drop rapidly, suggesting that the optimization-based embeddings in $\mathbb{H}_{\mathbb{R}}^{20}$ are not effective enough for reconstructing trees of such scales.

In comparison, UnitBall and TreeRep achieve stable performances for larger trees. TreeRep learns a tree structure from the data as an intermediate step and then embeds the learned trees into the hyperbolic space using Sarkar's construction (Sarkar, 2011). When the input data is a tree, TreeRep exactly recovers the original tree structure. Figure 1 shows that UnitBall achieves comparable or even better performances than TreeRep on the balanced trees. The results demonstrate that UnitBall does not compromise on trees. It produces high-quality embeddings for tree structures.

### 5.2.2 Results on compressed graphs

To illustrate the benefits of UnitBall on varying hierarchical structures, we now evaluate on the synthetic compressed graphs. The compressed graphs have local tree structures while being more

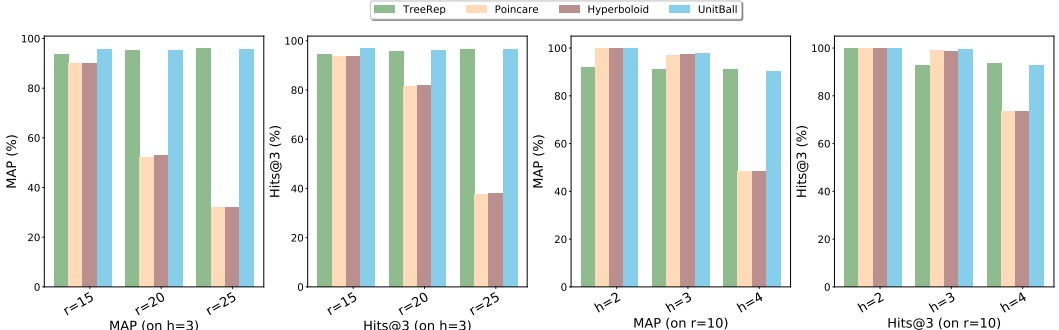

Figure 1: Evaluation of graph reconstruction on synthetic balanced trees in 20-d embedding spaces (10-d complex hyperbolic space for UnitBall). $r$ represents the degree while $h$ represents the depth.

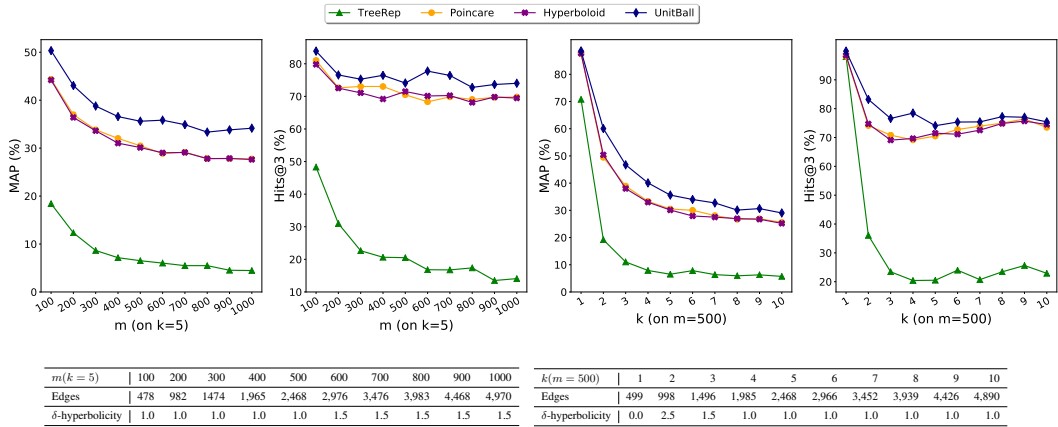

| $m(k=5)$ | 100 | 200 | 300 | 400 | 500 | 600 | 700 | 800 | 900 | 1000 |
|---|---|---|---|---|---|---|---|---|---|---|
| Edges | 478 | 982 | 1474 | 1,965 | 2,468 | 2,976 | 3,476 | 3,983 | 4,468 | 4,970 |
| $\delta$-hyperbolicity | 1.0 | 1.0 | 1.0 | 1.0 | 1.0 | 1.5 | 1.5 | 1.5 | 1.5 | 1.5 |

| $k(m=500)$ | 1 | 2 | 3 | 4 | 5 | 6 | 7 | 8 | 9 | 10 |
|---|---|---|---|---|---|---|---|---|---|---|
| Edges | 499 | 998 | 1,496 | 1,985 | 2,468 | 2,966 | 3,452 | 3,939 | 4,426 | 4,890 |
| $\delta$-hyperbolicity | 0.0 | 2.5 | 1.5 | 1.0 | 1.0 | 1.0 | 1.0 | 1.0 | 1.0 | 1.0 |

Figure 2: Evaluation of graph reconstruction on synthetic compressed graphs in 20-d embedding spaces (10-d complex hyperbolic space for UnitBall). $m$ represents the number of nodes in the graph while $k$ represents the number of random trees aggregated to the graph ($k$ controls the denseness and noise level of the graph). The statistics of the compressed graphs are provided in the tables.

complicated than trees. Each compressed graph-$(m, k)$ consists of $m$ nodes and is aggregated from $k$ random trees on the $m$ nodes. The bigger $k$ corresponds to the denser and noisier graph.

Figure 2 depicts the reconstruction results as a function of varying $m$ and $k$. The results on the compressed graphs are not as good as on balanced trees, especially with the increase of $m$ and $k$, which represents the increase of graph scale and denseness respectively. Notably, UnitBall outperforms all other methods on the challenging data, showing that UnitBall handles the noisy locally tree-like structures better. TreeRep has comparable results with other methods when $(m, k) = (500, 1)$ since when $k = 1$, the graph is exactly a tree, i.e., $\delta = 0$. However, when $k > 1$ and $\delta > 0$, TreeRep cannot achieve promising results, because when the data metrics deviate from tree metrics, it does not help much to learn a tree structure from the data as an intermediate step.

## 5.3 Link prediction

### 5.3.1 Overall results

In this section, we evaluate the performances on the link prediction task for the real-world taxonomies. Table 2 presents the results in 32-d embedding spaces for baselines and 16-d complex hyperbolic space for UnitBall. Predicting missing links requires stronger generalization capacity than reconstructing graphs, and UnitBall still has the best performances on all three datasets. Besides, we see that Euclidean shows shortages on these hierarchically-structured data, which is consistent with the results in previous works (Nickel and Kiela, 2017, 2018). Similar to the results on the graph reconstruction task, Poincaré and Hyperboloid have very close performances, while Hyperboloid has slightly better results. They have significant improvements over Euclidean, but they still fall behind UnitBall, which

Table 2: Evaluation of taxonomy link prediction in 32-d embedding spaces (16-d complex hyperbolic space for UnitBall). The best results are shown in boldface. The second best results are underlined.

| | ICD10 | | | YAGO3-wikiObjects | | | WordNet-noun | | |
|---|---|---|---|---|---|---|---|---|---|
| | MAP | MRR | Hits@3 | MAP | MRR | Hits@3 | MAP | MRR | Hits@3 |
| Euclidean | 3.75 | 3.72 | 2.39 | 4.85 | 4.45 | 2.78 | 5.59 | 5.36 | 3.16 |
| TreeRep | 4.96 | 7.92 | 8.49 | 20.19 | 21.85 | 27.19 | 9.30 | 9.98 | 11.90 |
| Poincaré | 35.24 | 34.45 | 52.71 | 30.06 | 28.47 | 41.61 | 25.46 | 23.99 | 27.80 |
| Hyperboloid | 34.80 | 34.01 | 52.88 | 30.80 | 29.21 | 43.17 | 25.65 | 24.15 | 27.50 |
| UnitBall | **47.88** | **46.96** | **70.28** | **33.33** | **31.85** | **47.41** | **27.29** | **25.93** | **32.95** |

Table 3: Evaluation of taxonomy link prediction in different embedding dimensions (the embedding dimension for UnitBall is half of other models). The best results are shown in boldface. The second best results are underlined.

| | YAGO3-wikiObjects | | | | | | | | |
|---|---|---|---|---|---|---|---|---|---|
| | 8-dimensional | | | 32-dimensional | | | 128-dimensional | | |
| | MAP | MRR | Hits@3 | MAP | MRR | Hits@3 | MAP | MRR | Hits@3 |
| Euclidean | 1.02 | 0.92 | 0.57 | 4.85 | 4.45 | 2.78 | 16.67 | 15.76 | 15.97 |
| TreeRep | 16.91 | 17.48 | 27.53 | 20.19 | 21.85 | 27.19 | 21.18 | 23.44 | 32.84 |
| Poincaré | 29.70 | 28.13 | 41.64 | 30.06 | 28.47 | 41.61 | 29.93 | 28.35 | 41.53 |
| Hyperboloid | 30.87 | 29.28 | 43.50 | 30.80 | 29.21 | 43.17 | 30.68 | 29.07 | 42.86 |
| UnitBall | **31.40** | **29.98** | **44.25** | **33.33** | **31.85** | **47.41** | **32.76** | **31.28** | **46.25** |

demonstrates our claims that the non-constant negative curvature of the complex hyperbolic space addresses the varying hierarchical structures on real-world datasets.

We notice that TreeRep does not perform well on the link prediction task. As mentioned in Section 2, the combinatorial construction-based embedding methods (Sala et al., 2018; Gu et al., 2019; Sonthalia and Gilbert, 2020) target on minimizing the reconstruction distortion of data and they can achieve very good results on the graph reconstruction task. But minimizing the reconstruction distortion may overfit the training set, thus resulting in the unpromising generalization performance for unobserved edges. Hence, they are more suitable to learn the representation of graph data without missing links. We also evaluate TreeRep on the real-world taxonomy reconstruction task in Appendix D.5.

### 5.3.2 Exploring the embedding dimensions

In this section, we explore the performances in different embedding dimensions. The results on YAGO3-wikiObjects are presented in Table 3. Results on other datasets are in Appendix D.6. We find that with the increase of the embedding dimension, Euclidean can have big improvements, but its performances in 128-d still cannot surpass other methods in 8-d. TreeRep also achieves better results with the increase of dimension, but overall its performances on the link prediction task are not very promising. By comparison, Poincaré, Hyperboloid, and UnitBall achieve great results steadily. 8-d is already enough for Poincaré and Hyperboloid to handle the link prediction task. We notice that UnitBall has small improvements from 4-d to 16-d, then converges to the stable performance. The results demonstrate that the Euclidean embeddings need to increase the dimension to better model the increasing complex hierarchies, while the complex hyperbolic space and the hyperbolic space have strong generalization competence for hierarchical structures.

## 6 Conclusion

In this paper, we present a novel approach for learning the embeddings of hierarchical structures in the unit ball model of the complex hyperbolic space. We characterize the geometrical properties of the complex hyperbolic space, including the variable negative curvature and the exponential growth of volume of geodesic balls, which are beneficial for data with various hierarchical structures. We exemplify the superiority of our approach over the graph reconstruction task and the link prediction task on both synthetic and real-world data, which cover the tree structures as well as the general hierarchical structures. The empirical results show that our approach outperforms the hyperbolic embedding methods in terms of representation capacity and generalization performance.

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
