# Appendix of Unit Ball Model for Embedding Hierarchical Structures in the Complex Hyperbolic Space

## A Proof of Theorem 1

In Section 3.3 in the paper, we presented Theorem 1 about the curvature of the complex hyperbolic space $\mathbb{H}^n_{\mathbb{C}}$ (Goldman, 1999):

**Theorem 1.** *The curvature is not constant in $\mathbb{H}^n_{\mathbb{C}}$. It is pinched between $-1$ (in the directions of complex projective lines) and $-1/4$ (in the directions of totally real planes).*

The sketch explanation is that all unit tangent vectors are equivalent, but not all directions are spanned by two unit tangent vectors. Before proving Theorem 1, we need to introduce the definition of *Kähler structure* (Mok, 1989).

**Definition 1** (Kähler structure)**.** *A **Kähler structure** can be defined in any of the following equivalent ways:*

 1. *A complex structure with a closed, positive $(1,1)$-form.*

 2. *A Riemannian structure with a complex structure such that the corresponding exterior 2-form is closed.*

 3. *A symplectic structure with a compatible integrable almost complex structure which is positive.*

Recall that in Section 3.3, we defined the complex hyperbolic space $\mathbb{H}^n_{\mathbb{C}}$ using the projectivization of the negative zone with a Hermitian form $\langle\!\langle \mathbf{z}, \mathbf{w} \rangle\!\rangle$. Denote $\omega$ as the imaginary part of the Hermitian form $\langle\!\langle , \rangle\!\rangle$, i.e., $\omega(\mathbf{z}, \mathbf{w}) = \frac{1}{2i}(\langle\!\langle \mathbf{z}, \mathbf{w} \rangle\!\rangle - \langle\!\langle \mathbf{w}, \mathbf{z} \rangle\!\rangle)$, then according to (Goldman, 1999), the metric $\omega$ is *positive* and *closed*, and necesssarily has type $(1,1)$. Then by the first definition in Definition 1, $\mathbb{H}^n_{\mathbb{C}}$ is a Kähler structure.

Let $M$ be a Kähler manifold and $\mathbf{z} \in M$. Denote $\mathcal{T}_{\mathbf{z}}M$ as the tangent space of $M$ at $\mathbf{z}$ and $J : \mathcal{T}M \to \mathcal{T}M$ is an endomorphism. As proved in (Kobayashi & Nomizu, 1963), the curvature of real 2-planes in the tangent space $\mathcal{T}_{\mathbf{z}}M$ has the following properties:

**Theorem 2.** *Let $M$ be a connected Kähler manifold of complex dimension $n \geq 2$. If the holomorphic sectional curvature $K(p)$, where $p$ is a plane in $\mathcal{T}_{\mathbf{z}}M$ invariant by $J$, depends only on $\mathbf{z}$, then $M$ is a space of constant holomorphic sectional curvature.*

Next, we give a proposition in (Kobayashi & Nomizu, 1963), which is about the curvature of a plane.

**Proposition 1.** *If $\mathbf{u}, \mathbf{v}$ is an orthonormal basis for a plane $p$ and if we set the curvature of $p$ as $K(p) = R(\mathbf{u}, \mathbf{v})$, where $R(\mathbf{u}, \mathbf{v})$ is the Riemann curvature tensor, then*

$$K(p) = \frac{1}{4}(1 + 3\cos^2 \alpha(p)),$$

*where $\alpha(p)$ is the angle between $p$ and $J(p)$.*

Submitted to 35th Conference on Neural Information Processing Systems (NeurIPS 2021). Do not distribute.

31 Finally, we prove Theorem 1 as follows.

32 *Proof.* Let $M$ be a Kähler manifold and $\mathbf{z} \in M$. From Theorem 2, the corresponding sectional
33 curvature function of real 2-planes in $\mathcal{T}_{\mathbf{z}}M$ is completely determined by the sectional curvature
34 function restricted to complex lines in $\mathcal{T}_{\mathbf{z}}M$. If the sectional curvature of every complex line in $\mathcal{T}M$
35 equals $\kappa$, then $M$ is said to have constant holomorphic sectional curvature $\kappa$.

36 Then from Proposition 1, we can know that in this case, the sectional curvature of a 2-dimensional
37 subspace $S \subset \mathcal{T}M$ is

$$K(S) = \kappa \frac{1 + 3\cos^2\alpha(S)}{4}, \tag{A.1}$$

38 where $\alpha(S)$ is the angle of holomorphy, defined as the smallest angle between two nonzero vectors
39 from two linear subspaces of the underlying real vector space of $M$.

40 In particular, the complex hyperbolic space $\mathbb{H}_{\mathbb{C}}^n$ is a Kähler structure with $\kappa = -1$. Since $0 \leq$
41 $\cos^2\alpha(S) \leq 1$, then from Eq. (A.1), we can have $-1 \leq K(S) \leq -1/4$ for any 2-dimensional
42 subspace $S \subset \mathcal{T}M$ of $\mathbb{H}_{\mathbb{C}}^n$, i.e., the (sectional) curvature is not constant in $\mathbb{H}_{\mathbb{C}}^n$, but pinched between
43 $-1$ and $-1/4$. Thus we proved the non-constant curvature of $\mathbb{H}_{\mathbb{C}}^n$.

44 Specifically, we discuss the complex projective lines and totally real planes in the unit ball model of
45 the complex hyperbolic space:

$$\mathcal{B}_{\mathbb{C}}^n = \{(z_1, \cdots, z_n, 1) | |z_1|^2 + \cdots + |z_n|^2 < 1\}. \tag{A.2}$$

46 First let's consider the case of complex projective lines. Consider a complex line $L$ in $\mathbb{C}^n$ that
47 intersects the unit ball model $\mathcal{B}_{\mathbb{C}}^n$. Let $\mathbf{z}$ be any point in $L \cap \mathcal{B}_{\mathbb{C}}^n$. We can apply an element of $\text{PU}(n, 1)$
48 to $L$ so that it becomes the last coordinate axis $\{(\mathbf{0}, z_n) | z_n \in \mathbb{C}\}$, whose intersection with $\mathcal{B}_{\mathbb{C}}^n$ is the
49 disk $|z_n| < 1$. Then the restriction of the Bergman metric to this disc is the Poincaré metric (Beardon,
50 2012) of constant curvature $-1$.

51 In order to see this, let $\mathbf{z} = (\mathbf{0}, z_n, 1)$ and $\mathbf{w} = (\mathbf{0}, w_n, 1)$, $\mathbf{z}, \mathbf{w} \in L \cap \mathcal{B}_{\mathbb{C}}^n$, then from Eq. (9) in
52 Section 4.1, the distance between $\mathbf{z}$ and $\mathbf{w}$ is given by

$$d_{\mathcal{B}_{\mathbb{C}}^n}(\mathbf{z}, \mathbf{w}) = arcosh(2\frac{\langle\!\langle \mathbf{z}, \mathbf{w} \rangle\!\rangle \langle\!\langle \mathbf{w}, \mathbf{z} \rangle\!\rangle}{\langle\!\langle \mathbf{z}, \mathbf{z} \rangle\!\rangle \langle\!\langle \mathbf{w}, \mathbf{w} \rangle\!\rangle} - 1), \tag{A.3}$$

53 where the Hermitian form $\langle\!\langle \mathbf{z}, \mathbf{w} \rangle\!\rangle$ is a standard Hermitian form:

$$\langle\!\langle \mathbf{z}, \mathbf{w} \rangle\!\rangle = z_1\overline{w_1} + \cdots + z_n\overline{w_n} - z_{n+1}\overline{w_{n+1}}. \tag{A.4}$$

54 Then we have

$$\cosh^2(\frac{d_{\mathcal{B}_{\mathbb{C}}^n}(\mathbf{z}, \mathbf{w})}{2}) = \frac{\langle\!\langle \mathbf{z}, \mathbf{w} \rangle\!\rangle \langle\!\langle \mathbf{w}, \mathbf{z} \rangle\!\rangle}{\langle\!\langle \mathbf{z}, \mathbf{z} \rangle\!\rangle \langle\!\langle \mathbf{w}, \mathbf{w} \rangle\!\rangle} = \frac{|z_n\overline{w_n} - 1|^2}{(|z_n|^2 - 1)(|w_n|^2 - 1)}, \tag{A.5}$$

55 which is just the Poincaré metric (Beardon, 2012).

56 Next consider a totally real plane $p$. Any totally real plane $p$ is the image under an element of
57 $\text{PU}(n, 1)$ of the subspace comprising those points of $\mathcal{B}_{\mathbb{C}}^n$ with real coordinates, that is actually an
58 embedded copy of the real hyperbolic space $\mathbb{H}_{\mathbb{R}}^n = \{(x_1, \ldots, x_n) | x_1, \ldots, x_n \in \mathbb{R}\}$. This subspace
59 intersects $\mathcal{B}_{\mathbb{C}}^n$ in the subset consisting of those points with $x_1^2 + \cdots + x_n^2 < 1$. Then the Bergman
60 metric restricted to this real-space unit ball is just the Klein-Beltrami metric (Ratcliffe et al., 1994) on
61 the unit ball in $\mathbb{R}^n$ with constant curvature $-1/4$.

62 To see this, let $\mathbf{x} = (x_1, \ldots, x_n, 1)$ and $\mathbf{y} = (y_1, \ldots, y_n, 1)$, $\mathbf{x}, \mathbf{y} \in \mathbb{H}_{\mathbb{R}}^n \cap \mathcal{B}_{\mathbb{C}}^n$, then apply the similar
63 process with the above, we have

$$\cosh^2(\frac{d_{\mathcal{B}_{\mathbb{C}}^n}(\mathbf{x}, \mathbf{y})}{2}) = \frac{\langle\!\langle \mathbf{x}, \mathbf{y} \rangle\!\rangle \langle\!\langle \mathbf{y}, \mathbf{x} \rangle\!\rangle}{\langle\!\langle \mathbf{x}, \mathbf{x} \rangle\!\rangle \langle\!\langle \mathbf{y}, \mathbf{y} \rangle\!\rangle} = \frac{(x_1y_1 + \cdots + x_ny_n - 1)^2}{(x_1^2 + \cdots + x_n^2 - 1)(y_1^2 + \cdots + y_n^2 - 1)}, \tag{A.6}$$

64 which is the Klein-Beltrami metric (Ratcliffe et al., 1994) on the unit ball in $\mathbb{R}^n$ with constant
65 curvature $-1/4$.

66 Therefore, we proved that the curvature of $\mathbb{H}_{\mathbb{C}}^n$ is $-1$ in the directions of complex projective lines
67 while $-1/4$ in the directions of totally real planes. $\qquad\square$

# B  Derivation of distance gradient in the unit ball model

The distance function in the unit ball model is given by Eq. (A.3). We need to compute the distance gradient $\nabla_E d_{\mathcal{B}_\mathbb{C}^n}(\mathbf{z}, \mathbf{w}) = \frac{\partial d_{\mathcal{B}_\mathbb{C}^n}(\mathbf{z},\mathbf{w})}{\partial \mathbf{x}} + i \frac{\partial d_{\mathcal{B}_\mathbb{C}^n}(\mathbf{z},\mathbf{w})}{\partial \mathbf{y}}$ during the Riemannian optimization. The full derivation is as follows.

First, we need to introduce Wirtinger derivatives (Wirtinger, 1927), which constructs a differential calculus for differential functions on complex domains.

**Definition 2** (Wirtinger derivatives). *The partial derivatives of a (complex) function $f(z)$ of a complex variable $z = x + iy \in \mathbb{C}, x, y \in \mathbb{R}$, with respect to $z$ and $\bar{z} = x - iy$, respectively, are defined as:*

$$\frac{\partial f(z, \bar{z})}{\partial z} = \frac{1}{2}\Big(\frac{\partial}{\partial x} - i\frac{\partial}{\partial y}\Big) f(z, \bar{z}), \qquad \frac{\partial f(z, \bar{z})}{\partial \bar{z}} = \frac{1}{2}\Big(\frac{\partial}{\partial x} + i\frac{\partial}{\partial y}\Big) f(z, \bar{z}).$$

The Wirtinger derivatives can be rewritten as:

$$\frac{\partial f(z, \bar{z})}{\partial x} = \Big(\frac{\partial}{\partial z} + \frac{\partial}{\partial \bar{z}}\Big) f(z, \bar{z}), \tag{B.1}$$

$$\frac{\partial f(z, \bar{z})}{\partial y} = i\Big(\frac{\partial}{\partial z} - \frac{\partial}{\partial \bar{z}}\Big) f(z, \bar{z}), \tag{B.2}$$

Let $p = \cosh(d_{\mathcal{B}_\mathbb{C}^n}(\mathbf{z}, \mathbf{w})) = 2\frac{\langle\!\langle \mathbf{z}, \mathbf{w}\rangle\!\rangle \langle\!\langle \mathbf{w}, \mathbf{z}\rangle\!\rangle}{\langle\!\langle \mathbf{z}, \mathbf{z}\rangle\!\rangle \langle\!\langle \mathbf{w}, \mathbf{w}\rangle\!\rangle} - 1$, then $d_{\mathcal{B}_\mathbb{C}^n}(\mathbf{z}, \mathbf{w}) = arcosh(p) = \ln(p + \sqrt{p^2 - 1})$.

Let $\mathbf{z} = (z_1, \ldots, z_n, 1) \in \mathcal{B}_\mathbb{C}^n$, then

$$\frac{\partial d_{\mathcal{B}_\mathbb{C}^n}(\mathbf{z}, \mathbf{w})}{\partial z_j} = \frac{\partial d_{\mathcal{B}_\mathbb{C}^n}(\mathbf{z}, \mathbf{w})}{\partial p} \cdot \frac{\partial p}{\partial z_j} = \frac{1}{\sqrt{p^2 - 1}} \cdot \frac{\partial p}{\partial z_j}$$

$$= \frac{2}{\sqrt{p^2 - 1}} \cdot \frac{\partial \frac{(z_1 \overline{w_1} + \cdots + z_n \overline{w_n} - 1) \cdot \langle\!\langle \mathbf{w}, \mathbf{z}\rangle\!\rangle}{(z_1 \overline{z_1} + \cdots + z_n \overline{z_n} - 1) \cdot \langle\!\langle \mathbf{w}, \mathbf{w}\rangle\!\rangle}}{\partial z_j}$$

$$= \frac{2}{\sqrt{p^2 - 1}} \cdot \Big(\frac{\overline{w_j} \langle\!\langle \mathbf{w}, \mathbf{z}\rangle\!\rangle}{\langle\!\langle \mathbf{z}, \mathbf{z}\rangle\!\rangle \cdot \langle\!\langle \mathbf{w}, \mathbf{w}\rangle\!\rangle} - \frac{\overline{z_j} \langle\!\langle \mathbf{z}, \mathbf{w}\rangle\!\rangle \cdot \langle\!\langle \mathbf{w}, \mathbf{z}\rangle\!\rangle}{\langle\!\langle \mathbf{z}, \mathbf{z}\rangle\!\rangle^2 \cdot \langle\!\langle \mathbf{w}, \mathbf{w}\rangle\!\rangle}\Big), \tag{B.3}$$

for $1 \le j \le n$. Similarly, we can have

$$\frac{\partial d_{\mathcal{B}_\mathbb{C}^n}(\mathbf{z}, \mathbf{w})}{\partial \overline{z_j}} = \frac{2}{\sqrt{p^2 - 1}} \cdot \Big(\frac{w_j \langle\!\langle \mathbf{z}, \mathbf{w}\rangle\!\rangle}{\langle\!\langle \mathbf{z}, \mathbf{z}\rangle\!\rangle \cdot \langle\!\langle \mathbf{w}, \mathbf{w}\rangle\!\rangle} - \frac{z_j \langle\!\langle \mathbf{z}, \mathbf{w}\rangle\!\rangle \cdot \langle\!\langle \mathbf{w}, \mathbf{z}\rangle\!\rangle}{\langle\!\langle \mathbf{z}, \mathbf{z}\rangle\!\rangle^2 \cdot \langle\!\langle \mathbf{w}, \mathbf{w}\rangle\!\rangle}\Big). \tag{B.4}$$

Then by Eqs. (B.1), (B.3), and (B.4), we can get

$$\frac{\partial d_{\mathcal{B}_\mathbb{C}^n}}{\partial x_j} = \frac{\partial d_{\mathcal{B}_\mathbb{C}^n}(\mathbf{z}, \mathbf{w})}{\partial z_j} + \frac{\partial d_{\mathcal{B}_\mathbb{C}^n}(\mathbf{z}, \mathbf{w})}{\partial \overline{z_j}} = \frac{4}{\sqrt{p^2 - 1}} \Big(\frac{Re(\langle\!\langle \mathbf{z}, \mathbf{w}\rangle\!\rangle w_j)}{\langle\!\langle \mathbf{z}, \mathbf{z}\rangle\!\rangle \langle\!\langle \mathbf{w}, \mathbf{w}\rangle\!\rangle} - \frac{\langle\!\langle \mathbf{z}, \mathbf{w}\rangle\!\rangle \langle\!\langle \mathbf{w}, \mathbf{z}\rangle\!\rangle x_j}{\langle\!\langle \mathbf{z}, \mathbf{z}\rangle\!\rangle^2 \langle\!\langle \mathbf{w}, \mathbf{w}\rangle\!\rangle}\Big). \tag{B.5}$$

Similarly, by Eqs. (B.2), (B.3), and (B.4), we can get

$$\frac{\partial d_{\mathcal{B}_\mathbb{C}^n}}{\partial y_j} = i\Big(\frac{\partial d_{\mathcal{B}_\mathbb{C}^n}(\mathbf{z}, \mathbf{w})}{\partial z_j} - \frac{\partial d_{\mathcal{B}_\mathbb{C}^n}(\mathbf{z}, \mathbf{w})}{\partial \overline{z_j}}\Big) = \frac{4}{\sqrt{p^2 - 1}} \Big(\frac{Im(\langle\!\langle \mathbf{z}, \mathbf{w}\rangle\!\rangle w_j)}{\langle\!\langle \mathbf{z}, \mathbf{z}\rangle\!\rangle \langle\!\langle \mathbf{w}, \mathbf{w}\rangle\!\rangle} - \frac{\langle\!\langle \mathbf{z}, \mathbf{w}\rangle\!\rangle \langle\!\langle \mathbf{w}, \mathbf{z}\rangle\!\rangle y_j}{\langle\!\langle \mathbf{z}, \mathbf{z}\rangle\!\rangle^2 \langle\!\langle \mathbf{w}, \mathbf{w}\rangle\!\rangle}\Big), \tag{B.6}$$

where $Re(\cdot)$ and $Im(\cdot)$ denote the real and the imaginary part respectively. Then we can have

$$\frac{\partial d_{\mathcal{B}_\mathbb{C}^n}}{\partial \mathbf{x}} = \frac{4}{\sqrt{p^2 - 1}} \Big(\frac{Re(\langle\!\langle \mathbf{z}, \mathbf{w}\rangle\!\rangle \mathbf{w})}{\langle\!\langle \mathbf{z}, \mathbf{z}\rangle\!\rangle \langle\!\langle \mathbf{w}, \mathbf{w}\rangle\!\rangle} - \frac{\langle\!\langle \mathbf{z}, \mathbf{w}\rangle\!\rangle \langle\!\langle \mathbf{w}, \mathbf{z}\rangle\!\rangle \mathbf{x}}{\langle\!\langle \mathbf{z}, \mathbf{z}\rangle\!\rangle^2 \langle\!\langle \mathbf{w}, \mathbf{w}\rangle\!\rangle}\Big), \tag{B.7}$$

$$\frac{\partial d_{\mathcal{B}_\mathbb{C}^n}}{\partial \mathbf{y}} = \frac{4}{\sqrt{p^2 - 1}} \Big(\frac{Im(\langle\!\langle \mathbf{z}, \mathbf{w}\rangle\!\rangle \mathbf{w})}{\langle\!\langle \mathbf{z}, \mathbf{z}\rangle\!\rangle \langle\!\langle \mathbf{w}, \mathbf{w}\rangle\!\rangle} - \frac{\langle\!\langle \mathbf{z}, \mathbf{w}\rangle\!\rangle \langle\!\langle \mathbf{w}, \mathbf{z}\rangle\!\rangle \mathbf{y}}{\langle\!\langle \mathbf{z}, \mathbf{z}\rangle\!\rangle^2 \langle\!\langle \mathbf{w}, \mathbf{w}\rangle\!\rangle}\Big), \tag{B.8}$$

which are Eqs. (17) and (18) in Section 4.3 in the paper.

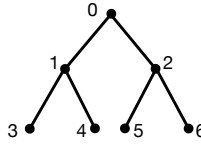 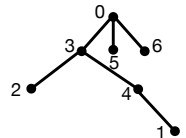 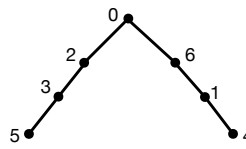 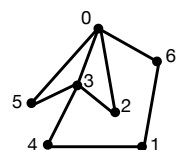

| balanced tree: r=2, h=2 | compressed graph: m=7, k=1 | compressed graph: m=7, k=1 | compressed graph: m=7, k=2 |
|---|---|---|---|
| $\delta$-hyperbolicity=0 | $\delta$-hyperbolicity=0 | $\delta$-hyperbolicity=0 | $\delta$-hyperbolicity=0.5 |

Figure 1: Simple examples of the synthetic data. The numbers $\{0, 1, \ldots, 6\}$ represent the nodes. The compressed graph-($m = 7$, $k = 2$) on the right are aggregated from the middle two compressed graphs-($m = 7$, $k = 1$).

## C  Definition of $\delta$-hyperbolicity

In this section, we give the definition of $\delta$-hyperbolicity (Gromov, 1987), which measures the tree-likeness of graphs. The lower $\delta$ corresponds to the more tree-like graph. Trees have 0 $\delta$-hyperbolicity.

**Definition 3** ($\delta$-hyperbolicity). *Let $a, b, c, d$ be vertices of the graph $G$. Let $S_1$, $S_2$ and $S_3$ be*

$$S_1 = dist(a,b) + dist(d,c), S_2 = dist(a,c) + dist(b,d), S_3 = dist(a,d) + dist(b,c).$$

*Suppose $M_1$ and $M_2$ are the two largest values among $S_1$, $S_2$, $S_3$ and $M_1 \geq M_2$. Define $hyp(a,b,c,d) = M_1 - M_2$. Then the $\delta$-**hyperbolicity** of $G$ is defined as*

$$\delta(G) = \frac{1}{2} \max_{a,b,c,d \in V(G)} hyp(a,b,c,d).$$

*That is, $\delta(G)$ is the maximum of $hyp$ over all possible 4-tuples $(a,b,c,d)$ divided by 2.*

## D  More experiments

### D.1  Synthetic data

In Section 5.1.1 in the paper, we introduced how we generate the synthetic data:

**Synthetic.** We generate various balanced trees and compressed graphs using the NetworkX package.[1] For **balanced trees**, we generate the balanced tree with degree $r$ and depth $h$. For **compressed graphs**, we generate $k$ random trees on $m$ nodes and then aggregate their edges to form a graph.

We give some examples of the synthetic data in Figure 1. As we can see, the compressed graphs-($m = 7$, $k = 1$) are random trees on 7 nodes, so their $\delta$-hyperbolicities are 0. The compressed graph-($m = 7$, $k = 2$) is no longer a tree after aggregating from two trees. Its local structures are more varying and complicated.

### D.2  Hardware

We conduct all the experiments except TreeRep on four NVIDIA GTX 1080Ti GPUs with 8GB memory each. For TreeRep, we need more memory to store the distance matrices, so we use a 96-core NVIDIA T4 GPU server with 503GB memory.

### D.3  Hyperparameters

For the baselines (TreeRep (Sonthalia & Gilbert, 2020),[2] Euclidean, Poincaré (Nickel & Kiela, 2017), and Hyperboloid (Nickel & Kiela, 2018)),[3] we use their public codes to train the embeddings. For all methods, we tune the hyperparameters by grid search. For the graph reconstruction task,

---

[1]https://networkx.org/documentation/stable/reference/generators.html

[2]https://github.com/rsonthal/TreeRep.

[3]https://github.com/facebookresearch/poincare-embeddings. The repository provides the implementation for Euclidean, Poincaré, and Hyperboloid.

Table 1: Hyperparameters of all methods.

| Model | Synthetic | ICD10 | YAGO3-wikiObjects | WordNet-noun |
|---|---|---|---|---|
| TreeRep | iterations: 20; optimization: *no opt*; pre-allocation fraction: 2.0; nthreads: 16; terminated edge weight: 0; trials/dataset: 3 | iterations: 32; optimization: *no opt*; pre-allocation fraction: 1.3; nthreads: 16; terminated edge weight: 0; trials/dataset: 3 | iterations: 32; optimization: *no opt*; pre-allocation fraction: 1.3; nthreads: 16; terminated edge weight: 0; trials/dataset: 3 | iterations: 1; optimization: *no opt*; pre-allocation fraction: 1.3; nthreads: 16; terminated edge weight: 0; trials/dataset: 3 |
| Euclidean | - | manifold: *euclidean*; learning rate: 1; epochs: 1500; dampening: 0.75; burnin: 20; burnin multiplier: 0.01; negative sample: 50; negative multiplier: 0.1; max norm: 50000 | manifold: *euclidean*; learning rate: 1; epochs: 1200; dampening: 0.75; burnin: 20; burnin multiplier: 0.01; negative sample: 50; negative multiplier: 0.1; max norm: 50000 | manifold: *euclidean*; learning rate: 1; epochs: 1500; dampening: 0.75; burnin: 20; burnin multiplier: 0.01; negative sample: 50; negative multiplier: 0.1; max norm: 50000 |
| Poincaré | manifold: *poincare*; learning rate: 0.3; epochs: 1500; dampening: 0.75; burnin: 20; burnin multiplier: 0.01; negative sample: 50; negative multiplier: 0.1; max norm: $1 - e^{-5}$ | manifold: *poincare*; learning rate: 1; epochs: 1500; dampening: 1.0; burnin: 20; burnin multiplier: 0.01; negative sample: 50; negative multiplier: 0.1; max norm: $1 - e^{-5}$ | manifold: *poincare*; learning rate: 1; epochs: 1200; dampening: 1.0; burnin: 20; burnin multiplier: 0.01; negative sample: 50; negative multiplier: 0.1; max norm: $1 - e^{-5}$ | manifold: *poincare*; learning rate: 1; epochs: 1500; dampening: 1.0; burnin: 20; burnin multiplier: 0.01; negative sample: 50; negative multiplier: 0.1; max norm: $1 - e^{-5}$ |
| Hyperboloid | manifold: *lorentz*; learning rate: 0.3; epochs: 1500; dampening: 0.75; burnin: 20; burnin multiplier: 0.01; negative sample: 50; negative multiplier: 0.1; max norm: *no-maxnorm* | manifold: *lorentz*; learning rate: 0.5; epochs: 1500; dampening: 1.0; burnin: 20; burnin multiplier: 0.01; negative sample: 50; negative multiplier: 0.1; max norm: *no-maxnorm* | manifold: *lorentz*; learning rate: 1; epochs: 1200; dampening: 1.0; burnin: 20; burnin multiplier: 0.01; negative sample: 50; negative multiplier: 0.1; max norm: *no-maxnorm* | manifold: *lorentz*; learning rate: 0.5; epochs: 1500; dampening: 1.0; burnin: 20; burnin multiplier: 0.01; negative sample: 50; negative multiplier: 0.1; max norm: *no-maxnorm* |
| UnitBall | manifold: *unitball*; learning rate: 8; epochs: 1500; dampening: 0.75; burnin: 20; burnin multiplier: 0.01; negative sample: 50; negative multiplier: 0.1; max norm: $1 - e^{-5}$ | manifold: *unitball*; learning rate: 11; epochs: 200; dampening: 1.0; burnin: 20; burnin multiplier: 0.01; negative sample: 50; negative multiplier: 0.1; max norm: $1 - e^{-5}$ | manifold: *unitball*; learning rate: 14; epochs: 1200; dampening: 1.0; burnin: 20; burnin multiplier: 0.01; negative sample: 50; negative multiplier: 0.1; max norm: $1 - e^{-5}$ | manifold: *unitball*; learning rate: 12; epochs: 900; dampening: 1.0; burnin: 20; burnin multiplier: 0.01; negative sample: 50; negative multiplier: 0.1; max norm: $1 - e^{-5}$ |

we tune the hyperparameters on balanced tree-(15,3) in 20-dimensional embedding spaces (10-dimensional complex hyperbolic space for UnitBall), while for the link prediction task, we tune the hyperparameters on the validation sets in 32-dimensional embedding spaces (16-dimensional complex hyperbolic space for UnitBall). The hyperparameters are given in Table 1.

## D.4 Evaluation

Our evaluation closely follows the setting of (Nickel & Kiela, 2017, 2018), which infers the hierarchies from distances in the embedding space. Specifically, for each test edge $(z, w)$, we compute the distance between the embeddings $d_{\mathcal{B}_{\mathbb{C}}^n}(\mathbf{z}, \mathbf{w})$ and rank it among the distances of all unobserved edges for $z$: $\{d_{\mathcal{B}_{\mathbb{C}}^n}(\mathbf{z}, \mathbf{w}') : (z, w') \notin \text{Training}\}$. We then report the following evaluation metrics of the rankings. Denote $E_{test}$ as the test edge set and $V = \{z | \exists w, (z, w) \in E_{test}\}$ as the test node set. Let $NE_z = \{w_1, w_2, \ldots, w_{|NE_z|}\}$ be the ground truth neighbor set of node $z$.

**Mean average precision (MAP).** The average precision (AP) is a way to summarize the precision-recall curve into a single value representing the average of all precisions and the MAP score is calculated by taking the mean AP over all classes. For a node $z$, from the learned embeddings, we can obtain the nodes closest to its embedding $\mathbf{z}$. Let $R_{z,w_i}$ be the smallest set of such nodes that contains $w_i$ (the $i$-th neighbor of $z$). Then the MAP is defined as:

$$\text{MAP} = \frac{1}{|V|} \sum_{z \in V} \frac{1}{|NE_z|} \sum_{w_i \in NE_z} Precision(R_{z,w_i}).$$

**Mean reciprocal rank (MRR).** The MRR is a statistic measure for evaluating a list of possible responses to a sample of queries, ordered by the probability of correctness. For a node $z$, from the learned embeddings, we can rank its distances with other nodes from the smallest to the largest. Let

Table 2: Evaluation of graph reconstruction on the real-world data (the dimension is 32 for TreeRep and 16 for UnitBall). For memory cost, the unit is *GiB*.

| | ICD10 | | | YAGO3-wikiObjects | | | WordNet-noun | | |
|---|---|---|---|---|---|---|---|---|---|
| | MRR | Hits@1 | Memory | MRR | Hits@1 | Memory | MRR | Hits@1 | Memory |
| TreeRep | 26.74 | 91.97 | 30 | 36.71 | 95.39 | 21 | 16.99 | 90.51 | 226 |
| UnitBall | 47.47 | 98.93 | 0.005 | 39.65 | 96.10 | 0.005 | 28.88 | 94.95 | 0.02 |

$rank_{w_i}$ be the rank of $w_i$ (the $i$-th neighbor of $z$). Then the MRR is defined as:

$$\text{MRR} = \frac{1}{|V|} \sum_{z \in V} \frac{1}{|NE_z|} \sum_{w_i \in NE_z} \frac{1}{rank_{w_i}}.$$

**The proportion of correct types that rank no larger than $N$ (Hits@$N$).** Hits@$N$ measures whether the top $N$ predictions contain the ground truth labels. For a node $z$, from the learned embeddings, we can obtain the set of $N$ nodes closest to its embedding $\mathbf{z}$, denoted as $R_z^N$. Then the Hits@$N$ is defined as:

$$\text{Hits@}N = \frac{1}{|V|} \sum_{z \in V} \mathbb{I}(|R_z^N \cap NE_z| \geq 1),$$

where $\mathbb{I}(|R_z^N \cap NE_z| \geq 1)$ is the indicator function.

### D.5    Comparison with TreeRep on real-world data reconstruction

In this section, we compare UnitBall with TreeRep on the real-world taxonomy reconstruction task. The results are presented in Table 2. As we analyzed in Section 5.3.1, TreeRep, as a combinatorial construction-based embedding method, is more suitable for the graph reconstruction task. Its performance is much better than that on the link prediction task. In addition, UnitBall outperforms TreeRep on reconstructing real-world taxonomies with varying structures.

We also notice the memory issues of the combinatorial construction-based embedding methods. Although TreeRep is very efficient in embedding tree structures since it does not need the gradient-based optimization steps, it costs more memory resources for constructing the tree structures from data. It is basically a computation time vs. memory cost trade-off issue. For a graph with $m$ nodes, TreeRep needs to construct a matrix of size $c \cdot m \times c \cdot m$ to construct the tree structure, where $1 \leq c \leq 2$ is a hyperparameter. We report the memory cost (*GiB*) in Table 2. UnitBall costs much less memory to learn the embeddings.

### D.6    More results on various dimensions

In Section 5.3.2, we reported the performances in different embedding dimensions on YAGO3-wikiObjects because of the page limits. Here we present the results in 8-d, 32-d, and 128-d embedding spaces (4-d, 16-d and 64-d complex hyperbolic spaces for UnitBall) on ICD10 and WordNet-noun in Table 3. Again, we see that with the increase of the embedding dimension, Euclidean can have big improvements, but its performances in 128-d still cannot surpass UnitBall and the hyperbolic models in 8-d. UnitBall outperforms the baselines almost all the time. Although on WordNet-noun, UnitBall in 4-d has slightly lower MAP and MRR than Poincaré and Hyperboloid in 8-d, it has significantly higher Hits@3.

### D.7    Comparison with trainable curvature method AttH

Our work focuses on the representation of single-relation graphs, which is a different research topic with multi-relational graph embeddings or knowledge graph embeddings, so it is hard to find an appropriate experimental setting to compare them. Nevertheless, to address the concerns of comparison with the trainable curvature method, here we evaluate AttH (Chami et al., 2020) on the single-relation taxonomy link prediction task. We tune the hyperparameters on the validation set and report the mean results over 5 running executions.

Table 3: Evaluation of taxonomy link prediction in different embedding dimensions (the embedding dimension for UnitBall is half of other models). The best results are shown in boldface. The second best results are underlined. TreeRep is not applicable to 128-d WordNet-noun due to the large memory cost so we do not include the results.

| | ICD10 | | | | | | | | |
| | 8-dimensional | | | 32-dimensional | | | 128-dimensional | | |
| | MAP | MRR | Hits@3 | MAP | MRR | Hits@3 | MAP | MRR | Hits@3 |
|---|---|---|---|---|---|---|---|---|---|
| Euclidean | 2.57 | 2.57 | 1.32 | 3.75 | 3.72 | 2.39 | 10.83 | 10.48 | 4.66 |
| TreeRep | 3.44 | 3.90 | 6.03 | 4.96 | 7.92 | 8.49 | 8.09 | 8.74 | 17.23 |
| Poincaré | 35.73 | 34.94 | 53.10 | 35.24 | 34.45 | 52.71 | 34.47 | 33.70 | 52.19 |
| Hyperboloid | 35.56 | 34.77 | 51.90 | 34.80 | 34.01 | 52.88 | 34.93 | 34.15 | 52.98 |
| UnitBall | **44.05** | **43.26** | **61.54** | **47.88** | **46.96** | **70.28** | **46.54** | **45.59** | **70.03** |
| | WordNet-noun | | | | | | | | |
| | 8-dimensional | | | 32-dimensional | | | 128-dimensional | | |
| | MAP | MRR | Hits@3 | MAP | MRR | Hits@3 | MAP | MRR | Hits@3 |
| Euclidean | 1.07 | 1.05 | 0.63 | 5.59 | 5.36 | 3.16 | 14.33 | 13.35 | 8.82 |
| Poincaré | 25.23 | 23.78 | 27.63 | 25.46 | 23.99 | 27.80 | 25.33 | 23.86 | 27.41 |
| Hyperboloid | **25.73** | **24.24** | 27.67 | 25.65 | 24.15 | 27.50 | 25.77 | 24.27 | 27.65 |
| UnitBall | 24.91 | 23.76 | **30.27** | **27.29** | **25.93** | **32.95** | **27.29** | **25.91** | **32.77** |

Table 4: Evaluation of taxonomy link prediction on YAGO3-wikiObjects (the dimension is 32 for AttH and 16 for UnitBall).

| | MAP | MRR | Hits@1 | Hits@3 |
|---|---|---|---|---|
| AttH | 30.22 | 28.47 | 9.10 | 43.83 |
| UnitBall | 33.33 | 31.85 | 15.62 | 47.41 |

From the results in Table 4, we see that UnitBall outperforms AttH in the single hypernymy relation link prediction task. However, UnitBall cannot infer multiple relations like AttH for now. Motivated by our theoretical grounding and empirical success, we believe the future work of the complex hyperbolic embeddings will have promising improvements on multi-relational graph embeddings.