# OpenReview forum: "Unit Ball Model for Embedding Hierarchical Structures in the Complex Hyperbolic Space"
_NeurIPS.cc/2021/Conference — NeurIPS 2021 Submitted_

### Official Review · Reviewer_XZ2r · 2021-07-15

**Rating:** 6
**Confidence:** 3

**Summary:**

The paper proposes the complex hyperbolic space for embedding hierarchical data. The authors demonstrate the superiority of the proposed optimization approach for the tasks of graph reconstruction and link prediction.

**Limitations And Societal Impact:**

Yes.

**Main Review:**

Strengths:
- To the best of my knowledge, the proposed approach is novel.
- In the experiments, the complex hyperbolic space outperforms the baselines (Euclidean embeddings and several implementations of hyperbolic embeddings).

Weaknesses (more details below):
- The technical part of the paper is hard to follow (defining notation and more explanation would help).
- The motivation is not very clear: in which cases non-constant curvature is expected to help.
- Limited experimental evaluation (unclear whether the proposed approach would help in more practical tasks).

Regarding motivation, it is claimed that a non-constant negative curvature is beneficial. However, it is not discussed why. The reason why negative curvature is beneficial for tree structures is the exponential growth of balls’ volumes (Nickel and Kiela, 2017); the examples of structures that are suitable for positive/zero/negative curvature are also discussed in (Gu et al., 2019). Similarly, it would be useful to understand for which structures the complex hyperbolic space is beneficial and why. Experiments suggest that for the considered synthetic graphs the complex hyperbolic space is indeed useful, but without much explanation.

Regarding the technical part, I have to admit that I could not fully digest the theoretical part of the paper. In particular, there is little background explaining the technical differences between standard and complex hyperbolic spaces. Regarding the background on curvature, I would expect to see all notation being defined. Then, the notation \sim seems to be not conventionally used: for instance, in (1) some terms depending on n seem to be missing, and in (2) the same happens. Regarding the proofs, is it true that the proof of Theorem 1 is not new and is provided just for completeness? Similar question regarding the derivations in Section 4.1. Also, how the formula (5) is obtained?  Am I right that the main technical novelty is the derivation in Section 4.3?

Some comments regarding the experiments:
- In the first set of experiments, the authors first compare the ability of different spaces to embed balanced trees. Here, they make an interesting observation: the performance of Poincare and Hyperboloid drops with the size of a tree, while the proposed UnitBall and the baseline TreeRep work quite well. For the standard hyperbolic space, combinatorial optimization (TreeRep) outperforms gradient-based approaches (Poincare and Hyperboloid). So, it is interesting to see that UnitBall works well while it also uses Riemannian SGD. In other words, we see that both complex and standard hyperbolic spaces work well, so what is the reason for UnitBall to outperform Poincare and Hyperboloid?
- The experiments are conducted on three real datasets. Comparing with (Nickel and Kiela, 2017), there is the same WordNet dataset, but the values for MAP are sufficiently smaller. What difference in the experimental setup causes this? It would be useful to see the numbers in a comparable setup.
- Unfortunately, the source code is not present.

Minor comments:
- \cite -> \citet in multiple places, when citations are used as part of the sentence (line 56 and below).
- Line 137 We -> we
- What is \mathcal{T} in line 156?

**Time Spent Reviewing:**

9

---

> ### Author Response · Authors · 2021-08-10
> **Response to Reviewer XZ2r (Part 1: Theoretical background)**
>
> We would like to thank the reviewer for the valuable comments.
>
> We would first like to emphasize that one of the main contributions of our paper is the proposition of the novel embedding approach defined in the unit ball model of complex hyperbolic space to handle nosier and more flexible hierarchies. Neither of the complex hyperbolic embeddings or the complex RSGD formulation has been defined or proposed before.
>
> With regard to the theoretical part, we would like to explain as follows to clear up the reviewer's confusion and misunderstanding.
>
> >In particular, there is little background explaining the technical differences between standard and complex hyperbolic spaces.
>
> The necessary preliminaries of the hyperbolic geometry and complex hyperbolic geometry are given in Section 3. As we specified in Introduction and Section 3, one of the main differences between the real and the complex hyperbolic space is the constant/variable negative curvature, which is highly related to our work and we expect to be favorable for embedding various hierarchical structures, so we focus on this property of the two geometries. We would appreciate it if you could tell us what other aspects of technical differences should be specifically explained. We would like to make the revisions as we believe they would make the paper easier to understand.
>
> >Regarding the background on curvature, I would expect to see all notation being defined.
>
> We apologize for the lack of a detailed description of $\nabla$ in Definition 1 (curvature), which may have caused your confusion. $\nabla$ indicates the Levi-Civita connection, which is an affine connection that preserves the metric and is torsion-free. We will add the definitions of the affine connection and Levi-Civita connection in our revision.
>
> >Then, the notation $\sim$ seems to be not conventionally used: for instance, in (1) some terms depending on n seem to be missing, and in (2) the same happens.
>
> $\sim$ is a symbol that represents asymptotic equivalence. The asymptomatic symbol $\sim$ is conventional in mathematics such as in statistics and geometric analysis. Please note that in Eqs. (1) and (2), we are presenting the growth of the ball's volume **with its radius $\rho$**, so we keep the order containing $\rho$ to clearly see whether the volume grows exponentially or polynomially with $\rho$. We refer the reviewer and interested readers to Chapter 3 in (Goldman, 1999) for further understanding.
>
> >Regarding the proofs, is it true that the proof of Theorem 1 is not new and is provided just for completeness? Similar question regarding the derivations in Section 4.1.
>
> Although we proved Theorem 1 through Kähler structure by ourselves, we agree that Theorem 1 is well studied in the complex hyperbolic geometry works (Goldman, 1999; Parker, 2003) and we believe there are other ways to prove it. Since proving Theorem 1 is not the most essential part of our paper, we put the proof in the appendix and focused on our contribution to the embedding approach in the main body of the paper.
>
> The derivations of the unit ball model in Section 4.1 are not novel. Like the Poincaré ball embeddings (Nickel and Kiela, 2017) and the hyperboloid embeddings  (Nickel and Kiela, 2018) which introduced the previously-established geometric models (the Poincaré ball model/the hyperboloid model) into the embedding algorithm and provided the required background of these models in their papers, we leverage the unit ball model to learn embeddings and give the required background such as the model's formula, metric, as well as distance function in Section 4.1.
>
> >Also, how the formula (5) is obtained?
>
> Eq. (5) is used to explain that complex hyperbolic space also has the tree-like exponential volume growth property. The derivation of Eq. (5) needs much page space and the derivation process is nonessential for our paper. We refer the reviewer and the interested readers to pg.104-105 in (Goldman, 1999) for the derivation of Eq. (5). We would like to add the derivation in our appendix to avoid possible confusion.
>
> >Am I right that the main technical novelty is the derivation in Section 4.3?
>
> Although the basic formulas of complex hyperbolic geometry and the unit ball model are already established in the framework of geometric group theory, they have not been applied in graph embeddings or even machine learning algorithms before. We would like to emphasize that our work is the first to propose to embed the graphs into the unit ball model. Neither of the complex hyperbolic embeddings or the complex RSGD formulation has been defined or proposed before. The contribution of graph embedding approaches is not to create a novel geometric model but to use the existing geometric models to develop new learning algorithms. For example,  Nickel and Kiela (2017) did not propose the Poincaré ball model. Instead, they made use of the formulas and properties of the Poincaré ball model, which have been well studied in mathematical history. We believe the technical novelty lies in applying theoretical and abstract knowledge to machine learning tasks.
>
> **References**:
>
> W. M. Goldman. Complex hyperbolic geometry. Oxford University Press, 1999.
>
> M. Nickel and D. Kiela. Poincaré embeddings for learning hierarchical representations. In NIPS, pages 6338–6347, 2017.
>
> M. Nickel and D. Kiela. Learning continuous hierarchies in the lorentz model of hyperbolic geometry. In ICML, volume 80 of Proceedings of Machine Learning Research, pages 3776–3785. PMLR, 2018.
>
> J. R. Parker. Notes on complex hyperbolic geometry. preprint, 2003. URL: https://maths.dur.ac.uk/~dma0jrp/img/NCHG.pdf.

---

> ### Author Response · Authors · 2021-08-10
> **Response to Reviewer XZ2r (Part 2: Motivation and experiments)**
>
> We address the comments on motivation and experiments as follows:
>
> >Regarding motivation, it is claimed that a non-constant negative curvature is beneficial. However, it is not discussed why. The reason why negative curvature is beneficial for tree structures is the exponential growth of balls’ volumes (Nickel and Kiela, 2017); the examples of structures that are suitable for positive/zero/negative curvature are also discussed in (Gu et al., 2019). Similarly, it would be useful to understand for which structures the complex hyperbolic space is beneficial and why. Experiments suggest that for the considered synthetic graphs the complex hyperbolic space is indeed useful, but without much explanation.
>
> We would like to emphasize that our motivation is to make use of the variable negative curvature to flexibly handle the varying hierarchical structures. Instead of tackling specific graphs with specific curvatures, we learn the embeddings directly in the complex hyperbolic space. The learned embeddings are located in different submanifolds of the unit ball model, whose curvatures are different. We expect the learning process to implicitly align the geometric structures and the underlying graph structures since the loss function aims at preserving the hierarchical relationships among nodes. The experiments on the synthetic compressed graphs and the real-world taxonomies demonstrate the unit ball embeddings indeed capture the varying hierarchical structures better.
>
> We would like to explain more about the synthetic graphs. For the experiments on balanced trees, our goal is to show that UnitBall does not compromise on trees. The reason is that the complex hyperbolic space maintains the exponential volume growth property as addressed in Section 3.3. The experiments on compressed graphs are used to demonstrate the advantages of UnitBall in various hierarchically structured data. Figure 2 in the paper shows the outperformance of the unit ball embeddings and provides the $\delta$-hyperbolicity of the synthetic graphs. With the increase of k, the graph becomes denser and more varying-structured. The real hyperbolic embedding methods relying on the constant negative curvature fail to capture these graphs which are aggregated from random trees and contain multitree structures, while UnitBall outperforms the baselines a lot, revealing that learning embeddings in the complex hyperbolic space with variable negative curvature handles the noisy locally tree-like structures better.
>
> >unclear whether the proposed approach would help in more practical tasks.
>
> As stated in the Introduction, the complex hyperbolic embedding approach targets the representation learning of data with hierarchical structures, which is an important machine learning task with many applications, such as taxonomy induction and hypernymy detection. In our experiments, we used the proposed approach to solve the graph reconstruction task and link prediction task and we dealt with the real-world taxonomies.
>
> The future work of complex hyperbolic embeddings includes complex hyperbolic knowledge graph (KG) embeddings and applications in neural networks. It is known that the Poincaré embeddings inspired the hyperbolic KG embeddings and hyperbolic neural networks (these related works are listed in Section 2 in the paper), which achieved promising performances in many downstream tasks. Motivated by our theoretical grounding and empirical success, we believe the complex hyperbolic embeddings will help to improve the existing KG embedding works, graph neural networks, and other related applications.
>
> >In the first set of experiments, the authors first compare the ability of different spaces to embed balanced trees. Here, they make an interesting observation: the performance of Poincare and Hyperboloid drops with the size of a tree, while the proposed UnitBall and the baseline TreeRep work quite well. For the standard hyperbolic space, combinatorial optimization (TreeRep) outperforms gradient-based approaches (Poincare and Hyperboloid). So, it is interesting to see that UnitBall works well while it also uses Riemannian SGD. In other words, we see that both complex and standard hyperbolic spaces work well, so what is the reason for UnitBall to outperform Poincare and Hyperboloid?
>
> We think the reason for Unitball to outperform Poincaré and Hyperboloid is that the complex hyperbolic space has more capacity than the real hyperbolic space. Intuitively, a complex hyperbolic embedding vector has a real part and an imaginary part, which are constrained to be within the unit ball model instead of simply the product of two real embeddings or a real embedding with double dimension.
>
> The reviewer also mentioned the good performance of TreeRep in the first experiment. As explained in Section 5.2, TreeRep learns a tree structure from the data as an intermediate step and then embeds the learned trees into the hyperbolic space using Sarkar’s construction. When the input graph is a tree (the case in the balanced trees), TreeRep exactly recovers the original tree structure. In addition, we discussed in Section 5.3 that the combinatorial construction-based embedding methods target minimizing the reconstruction distortion of data, so they can achieve very good results on the graph reconstruction task but compromise on link prediction.
>
> >Comparing with (Nickel and Kiela, 2017), there is the same WordNet dataset, but the values for MAP are sufficiently smaller. What difference in the experimental setup causes this? It would be useful to see the numbers in a comparable setup.
>
> The original paper of (Nickel and Kiela, 2017) did not provide the information about data split for the link prediction task. In their Github repository, only the full transitive closure of WordNet is provided and there are no train-valid-test sets. Their released code works in the graph reconstruction way, i.e., the training set is the same as the test set. Therefore, we split the WordNet dataset by ourselves and train the Poincaré embeddings using the training set while evaluating the test set. The details of our data splitting and experimental settings can be referred to Section 5.1 and Appendix D.
>
> >Unfortunately, the source code is not present.
>
> As addressed in the paper, the code is proprietary for this moment. The code and data will be publicly released soon after the publication of the paper.
>
> >Minor comments.
>
> Thanks for giving the edit suggestions. $\mathcal{T}$ in Line 156 is a redundant typo that we had intended to represent the training set and should be deleted. We will make the revisions in our final version.
>
> **References**:
>
> M. Nickel and D. Kiela. Poincaré embeddings for learning hierarchical representations. In NIPS, pages 6338–6347, 2017.

---

> ### Author Response · Authors · 2021-08-25
> **Looking forward to further discussions**
>
> We greatly appreciate your valuable comments and feedback. In our response, we responded to your confusion on the theoretical background. We also addressed your concerns on motivation and experiments.
>
> Since there is one week left to continue discussion, we would like to know whether you have any concerns or questions after reading our response. We look forward to further discussions to address your comments.

---

### Official Review · Reviewer_BtXz · 2021-07-16

**Rating:** 6
**Confidence:** 4

**Summary:**

Complex hyperbolic space is new for embedding.

**Limitations And Societal Impact:**

The authors adequately addressed the limitations and potential negative societal impact of their work.

**Main Review:**

The work is technically sound and well organized. But there are some problems that I am concerned about:

(1) "The real-world hierarchically structured data are usually not trees since they can have varying local structures while being tree-like globally".  The authors present a novel embedding approach, which takes advantage of the variable negative curvature of the complex hyperbolic space, to handle data with complicated and various hierarchical structures.  However, it is not clear how the complex hyperbolic space solves the problem and why complex hyperbolic space can solve the problem of varying local structures.

(2) HGCN [1] leverages the trainable curvature to compensate for the disparity between the actual data structures and the constant-curvature hyperbolic space; the mixed curvature method [2] introduce a heuristic to estimate the sectional curvature of graph data and directly determine an appropriate signature; Graph Geometry Interaction Learning[3] also try to the problem of varying local structures? how about comparing it with this method?

References:

[1]NeurIPS'19 Hyperbolic graph neural network

[2]ICLR'19 Learning Mixed-Curvature Representations in Product Spaces

[3] NeurIPS'20 Graph Geometry Interaction Learning




**Time Spent Reviewing:**

8 Hours

---

> ### Author Response · Authors · 2021-08-10
> **Response to Reviewer BtXz**
>
> We would like to thank the reviewer for the valuable comments. We address the concerns as follows:
>
> >However, it is not clear how the complex hyperbolic space solves the problem and why complex hyperbolic space can solve the problem of varying local structures.
>
> We would like to emphasize that our motivation is to make use of the variable negative curvature to flexibly handle the varying hierarchical structures. Instead of tackling specific graphs with specific curvatures, we learn the embeddings directly in the complex hyperbolic space. The learned embeddings are located in different submanifolds of the unit ball model, whose curvatures are different. We expect the learning process to implicitly align the geometric structures and the underlying graph structures since the loss function aims at preserving the hierarchical relationships among nodes.
>
> We would like to take the experiments on the synthetic compressed graphs as an example. Figure 2 in the paper shows the outperformance of the unit ball embeddings and provides the $\delta$-hyperbolicity of the synthetic graphs. With the increase of k, the graph becomes denser and more varying-structured. The real hyperbolic embedding methods relying on the constant negative curvature fail to capture these graphs which are aggregated from random trees and contain multitree structures while UnitBall outperforms the baselines a lot, revealing that learning embeddings in the complex hyperbolic space with variable negative curvature handles the noisy locally tree-like structures better.
>
> >HGCN leverages the trainable curvature to compensate for the disparity between the actual data structures and the constant-curvature hyperbolic space; the mixed curvature method introduce a heuristic to estimate the sectional curvature of graph data and directly determine an appropriate signature; Graph Geometry Interaction Learning also try to the problem of varying local structures? how about comparing it with this method?
>
> 1. As we explained in Section 2, our work focuses on the single-relation graph embeddings and taxonomy embeddings, so we do not evaluate the multi-relational knowledge graph embedding models or the neural networks in our tasks. Although GNNs (including HGCN and GIL mentioned by the reviewer) also involve the graph embeddings and can deal with link prediction task, they make use of not only the edges between nodes but also the node features. The message propagation and attention mechanism make GNNs more flexible to handle various downstream tasks than shallow embeddings. However, the shallow embeddings usually motivate the improvements of the deep models. For instance, the Poincaré embeddings inspired the development of hyperbolic deep learning including HGCN.
> Nevertheless, to address the concerns of the reviewer, here we evaluate UnitBall on the link prediction task on the datasets of GIL. The results of HGCN and GIL are copied from Table 2 of the GIL's original paper (Zhu et al., 2020). We strictly follow their experimental settings and report the mean results of UnitBall in ROC AUC over 5 running executions. The dimension is 8 for UnitBall and 16 for HGCN and GIL.
> |          | Disease | Airport | Pubmed | Citeseer | Cora  |
> |----------|---------|---------|--------|----------|-------|
> | HGCN     | 90.80   | 96.43   | 95.13  | 96.63    | 93.81 |
> | GIL      | 99.90   | 98.77   | 95.49  | 99.85    | 98.28 |
> | UnitBall | 99.09   | 96.61   | 98.80  | 99.34    | 97.64 |
>
> We can see that UnitBall outperforms HGCN on the five datasets. GIL is slightly better than UnitBall on most datasets while being outperformed by UnitBall on Pubmed. The results are very promising for UnitBall since UnitBall is a shallow embedding approach without deep architecture or feature interaction. Motivated by our theoretical grounding and empirical success, we believe the complex hyperbolic embeddings will help to improve the GNNs and bring more insights into geometric deep learning.
>
> 2. As for the mixed curvature method in the product space, we discussed in Section 2 that it is impractical to search for the best manifold combination among enormous combinations for each new structure. We would like to add the remark that finding the accurate combination of manifolds costs too much search space even with the estimated signature since the number of factors in the signature is manually allocated. Moreover, TreeRep is the SOTA combinatorial construction-based hyperbolic embedding method, so among the construction-based baselines, we only compared with TreeRep in our experiments.
>
> **References**:
>
> S. Zhu, S. Pan, C. Zhou, J. Wu, Y. Cao, and B. Wang.  Graph geometry interaction learning.  In NeurIPS, 2020.

---

> > ### Comment · Reviewer_BtXz · 2021-08-26
> > **Thanks for your rebuttal**
> >
> > Thanks for your response and new experiments. The rebuttal has solved part of my concern, I raise my score from 5->6.
> >
> > But how could the learning process implicitly align the geometric structures and the underlying graph structures? is any special or different optimization process for the distance function of Complex Poincare?

---

> > > ### Author Response · Authors · 2021-08-30
> > > **Response to Reviewer BtXz**
> > >
> > > Thank you for your response. We greatly appreciate your questions and concerns.
> > > We now address your additional comments as follows:
> > >
> > > >But how could the learning process implicitly align the geometric structures and the underlying graph structures? is any special or different optimization process for the distance function of Complex Poincare?
> > >
> > > As introduced in Section 4.2 in the paper, the soft ranking loss Eq. (10) aims at preserving the hierarchical relationships in the graph. The underlying graph structures are reflected by the hierarchical relationships (i.e., the edges among nodes). Therefore, we optimize the loss to learn the graph embeddings which maintain the graph structures. The learned embeddings are located in different submanifolds of the unit ball model, whose curvatures and local geometric structures are different. The variable negative curvature of the complex hyperbolic space provides the capacity to deal with more varying structures.
> > >
> > > We would like to thank you again for your valuable comments and appreciate any  further questions.

---

> ### Author Response · Authors · 2021-08-25
> **Looking forward to further discussions**
>
> We greatly appreciate your valuable comments and feedback. In our response, we addressed your concerns on how the complex hyperbolic space with variable negative curvature can solve the problem of varying local structures. We also provided experimental results to address your concerns on comparison with HGCN and GIL.
>
> Since there is one week left to continue discussion, we would like to know whether you have any concerns or questions after reading our response. We look forward to further discussions to address your comments.

---

### Official Review · Reviewer_TUo7 · 2021-07-16

**Rating:** 5
**Confidence:** 4

**Summary:**

This paper proposes a method to learn embeddings in complex hyperbolic spaces. In contrast with real hyperbolic spaces, complex hyperbolic spaces have a variable negative curvature, which can handle more flexible graph structures. Experiments show that complex hyperbolic embeddings outperform real hyperbolic embeddings on synthetic and real-world graphs.

**Limitations And Societal Impact:**

Yes.

**Main Review:**

Strengths:
- This paper is interesting and the first to leverage complex hyperbolic spaces.
- The paper is well written and easy to follow.
- Experiments show significant improvements from using complex hyperbolic geometry.

Weaknesses:
- The main weakness of this work is the lack of motivation for using a complex hyperbolic space. While the authors mention that having a variable negative curvature can capture diverse structures, it is not clear how these variations in curvature relate to different degrees of hierarchy in graphs. It would be good to provide intuition for how curvature changes in this geometry, visualizations would have been very helpful here. Simple examples of graphs where this space is a better candidate than the real hyperbolic space would make the motivation clearer.

Minor comments:
- It would be interesting to understand the precision-distortion tradeoffs in complex hyperbolic geometry. For instance, it is known that real hyperbolic spaces are ideal candidates for balanced trees (section 5.2.1) but have limited capacity due to limited machine precision (Sala et al. 2018). It would therefore be interesting to investigate whether the improvements of complex hyperbolic embeddings are related to precision, and if distortion scales better with precision in this geometry.
- The Levi-Civita connection notation in definition 1 is not defined.
- Moving eq. (6) to section 3.3 would improve the flow.


**Time Spent Reviewing:**

2

---

> ### Author Response · Authors · 2021-08-10
> **Response to Reviewer TUo7**
>
> We would like to thank the reviewer for the valuable comments. We address the comments as follows:
>
> >The main weakness of this work is the lack of motivation for using a complex hyperbolic space. While the authors mention that having a variable negative curvature can capture diverse structures, it is not clear how these variations in curvature relate to different degrees of hierarchy in graphs. It would be good to provide intuition for how curvature changes in this geometry, visualizations would have been very helpful here. Simple examples of graphs where this space is a better candidate than the real hyperbolic space would make the motivation clearer.
>
> We appreciate the suggestion. However, curvature in the complex hyperbolic space is a very complicated topic in geometric group theory and differential geometry. The complex projective lines and the totally real planes are two kinds of special subspaces, whose curvatures are presented in Appendix A. For the subspace that lives in between the two special cases, its curvature can be computed accordingly. To the best of our knowledge, we could not find any prior work or research study that elaborated or visualized the curvature in the complex hyperbolic space everywhere. We refer the reviewer and readers to (N. Fisher) for an interesting example in the complex hyperbolic space (its curvature differs with our work with a constant multiplier 4). In Section 5 and Figure 5 of  (N. Fisher), the author explored the curvature of a triangle in complex hyperbolic geometry with numerical computation. Since digging into all the curvature details of the geometry is not the essential part of our work, we have omitted the related content in our paper.
>
> We would like to emphasize that our motivation is to make use of the variable negative curvature to flexibly handle the varying hierarchical structures. Instead of tackling specific graphs with specific curvatures, we learn the embeddings directly in the complex hyperbolic space. The learned embeddings are located in different submanifolds of the unit ball model, whose curvatures are different. We expect the learning process to implicitly align the geometric structures and the underlying graph structures since the loss function aims at preserving the hierarchical relationships among nodes.
>
> The experiments on the synthetic compressed graphs and the real-world taxonomies demonstrate the unit ball embeddings are indeed good at capturing the varying hierarchical structures. Note that the synthetic compressed graphs are simple examples of graphs. Figure 2 in the paper shows the outperformance of the unit ball embeddings and provides the $\delta$-hyperbolicity of the synthetic graphs. With the increase of m (representing graph scale) and k (representing the denseness and noisiness), UnitBall outperforms the baselines a lot, revealing that UnitBall handles the noisy locally tree-like structures better. Since we use the same loss function with Poincaré and Hyperboloid but learn in the unit ball model, the comparisons among UnitBall, Poincaré, Hyperboloid reveal the complex hyperbolic space is a better candidate than the real hyperbolic space.
>
> >Minor comments: It would be interesting to understand the precision-distortion tradeoffs in complex hyperbolic geometry. For instance, it is known that real hyperbolic spaces are ideal candidates for balanced trees (section 5.2.1) but have limited capacity due to limited machine precision (Sala et al. 2018). It would therefore be interesting to investigate whether the improvements of complex hyperbolic embeddings are related to precision, and if distortion scales better with precision in this geometry.
>
> For the experiments on balanced trees, our goal is to keep the compatible representational power with TreeRep since TreeRep exactly recovers the original tree structure when the input graph is a tree. Figure 1 shows that UnitBall achieves comparable or even better performances than TreeRep on the balanced trees. The results demonstrate that UnitBall does not compromise on trees. It produces high-quality embeddings for tree structures.
>
> We follow the experimental settings of previous works on optimization-based embeddings (Nickel and Kiela, 2017, 2018) and choose MAP, MRR, and Hits@N as metrics. We agree that distortion is another important evaluation metric for graph reconstruction task. The combinatorial construction-based methods directly minimize the distortion and evaluate the distortion. We expect the exploration on the combinatorial construction methods as well as the precision-distortion tradeoffs in complex hyperbolic geometry. That would be a challenging and interesting future work. We sincerely appreciate the suggestion.
>
> >The Levi-Civita connection notation in definition 1 is not defined.
>
> We apologize for the lack of a detailed description of $\nabla$ in Definition 1. $\nabla$ indicates the Levi-Civita connection, which is an affine connection that preserves the metric and is torsion-free. We will add the definitions of the affine connection and Levi-Civita connection in our revision.
>
> >Moving eq. (6) to section 3.3 would improve the flow.
>
> Eq. (6) is a specific example of the Hermitian form, which can be used to derive the unit ball model. There are other choices of the Hermitian form, which corresponds to other models of the complex hyperbolic geometry, such as the Siegel domain model. Since Section 3.3 introduces the complex hyperbolic geometry instead of the specific models of the geometry, we have not given the concrete Hermitian form in Section 3.3.
>
> **References**:
>
> N. Fisher. Notes on curvature in complex hyperbolic space. URL: https://sites.tufts.edu/natefisher/files/2020/11/Write-up.pdf.
>
> M. Nickel and D. Kiela.  Poincaré embeddings for learning hierarchical representations.  In NIPS, pages 6338–6347, 2017.
>
> M. Nickel and D. Kiela. Learning continuous hierarchies in the lorentz model of hyperbolic geometry. In ICML, volume 80 of Proceedings of Machine Learning Research, pages 3776–3785. PMLR, 2018.

---

> > ### Comment · Reviewer_TUo7 · 2021-09-01
> > **Thank you for your response**
> >
> > I thank the authors for their response. After reading the rebuttal, my main concern about the motivation of this work (point that was also raised by other reviewers) remains. It is still not clear which graphs are good candidates for the complex hyperbolic space. I looked through the appendices as well but did not find any clear explanation linking the graphs used in the paper to this geometry.
> >
> > I think that in order for this work to be useful in applications, it is crucial to better investigate which graphs are good candidates to be embedded in the complex hyperbolic space. I therefore updated my score to a weak reject, and encourage the authors to better motivate their work by investigating the benefits of this geometry more in depth.

---

> > > ### Author Response · Authors · 2021-09-02
> > > **Response to Reviewer TUo7**
> > >
> > > Thank you for your response. We appreciate your concern about the motivation of our work and your suggestion to improve our paper. We would like to explain more about our motivation.
> > >
> > > >It is still not clear which graphs are good candidates for the complex hyperbolic space. I looked through the appendices as well but did not find any clear explanation linking the graphs used in the paper to this geometry.
> > >
> > > Our motivation is to make use of the variable negative curvature of the complex hyperbolic space to flexibly handle the varying hierarchical structures. We understand that you want a clear elaboration about which kinds of graphs can benefit from the complex hyperbolic space. However, our work proposed an embedding algorithm that has strong capacities for the general graphs with hierarchical structures, instead of tackling specific graphs. Our claim of contributions and our experiments clearly focus on the data with complicated and various hierarchical structures. In our paper, we demonstrated the advantages of our approach on various hierarchically structured data, including synthetic graphs and real-world taxonomies.
> > >
> > > Please note that the claims of the hyperbolic embeddings  (Nickel and Kiela, 2017, 2018) were their contributions on hierarchical graphs. They also did not give specific candidates of the Poincaré ball/hyperboloid model. Because the hyperbolic models as well as our proposed complex hyperbolic model are not constrained to some narrower scope of hierarchical graphs. In our paper, we addressed the limitations of the hyperbolic embedding methods on hierarchical structures and proposed to use the complex hyperbolic space with variable curvature to solve the challenges. The empirical evaluation showed our approach had significant improvements over the hyperbolic embeddings.
> > >
> > > All graphs used in our paper were provided with their $\delta$-hyperbolicity, which measures the tree-likeness of graphs, reflecting the deviations from trees of the hierarchical graphs. The details of the synthetic graphs generation and the statistics of the real-world taxonomies were also given in the paper. We would appreciate it if you could tell us what other aspects of explanation linking the graphs should be specifically given. We would be glad to add these explanations as we believe they would make the paper easier to understand.
> > >
> > > >I think that in order for this work to be useful in applications, it is crucial to better investigate which graphs are good candidates to be embedded in the complex hyperbolic space.
> > >
> > > In addition to the above response on the 'good candidates' issue, we would like to address your concerns on the usefulness in applications. In our experiments, our approach performed very well on real-world taxonomies in the graph reconstruction task and the link prediction task. The experimental results in our response to Reviewer BtXz provided more promising results on various hierarchical graphs and even more general graphs. We find that all reviewers reached a consensus on our big improvements in experiments. Motivated by our theoretical grounding and empirical success, we believe the complex hyperbolic embeddings will help to improve the existing KG embedding works, graph neural networks, and other related applications.
> > >
> > > We would greatly appreciate it if you can reconsider your assessment of our work. We notice that you did not raise any new concern or question in your post-rebuttal response but maintaining the previous concern on motivation. We sincerely hope this response could address your concerns and confusion. Thank you again for your valuable comments and efforts to help to improve our paper.
> > >
> > > **References**:
> > >
> > > M. Nickel and D. Kiela. Poincaré embeddings for learning hierarchical representations. In NIPS, pages 6338–6347, 2017.
> > >
> > > M. Nickel and D. Kiela. Learning continuous hierarchies in the lorentz model of hyperbolic geometry. In ICML, volume 80 of Proceedings of Machine Learning Research, pages 3776–3785. PMLR, 2018.

---

> ### Author Response · Authors · 2021-08-25
> **Looking forward to further discussions**
>
> We greatly appreciate your valuable comments and feedback. In our response, we addressed your concerns on motivation for complex hyperbolic space. And we would like to revise our manuscript per your minor comments.
>
> Since there is one week left to continue discussion, we would like to know whether you have any concerns or questions after reading our response. We look forward to further discussions to address your comments.

---

### Decision · Program_Chairs · 2021-09-27

**Decision:**

Reject

**Comment:**

The paper proposes a new method to learn representations of hierarchical structures by embedding them into complex hyperbolic space. Reviewers and the AC agreed that the proposed approach is a promising direction to advance hyperbolic embeddings as it allows for variable negative curvature which has, for instance, the potential to improve the embedding quality of real-world graphs (which in almost all cases won't be represented adequately with constant curvature). Furthermore, the proposed approach shows promising empirical results and is clearly relevant to the NeurIPS community. However, all reviewers raised concerns regarding the clarity of discussion, especially with regard to the motivation and justification of the approach as well as the presentation of technical aspects. These concerns were not resolved after the author response and further discussion among reviewers. In relation to this, reviewers raised also concerns regarding the experimental evaluation as further analyses could provide needed insights into the approach and how it utilizes variable curvature. After rebuttal and discussion, no reviewer strongly supported the acceptance of the paper, with two reviewers indicating weak accept and one reviewer indicating weak reject. Taking rebuttal and discussion into account, I narrowly agree that the manuscript is currently not ready yet for acceptance at NeurIPS and would require an additional revision to resolve the above concerns. However, the idea to utilize complex hyperbolic space is clearly promising and I encourage the authors to revise and resubmit their work based on the feedback from this reviewing round.